# REVERSIBLE COLUMN NETWORKS

**Yuxuan Cai**[1]    **Yizhuang Zhou**[1]    **Qi Han**[1]    **Jianjian Sun**[1]    **Xiangwen Kong**[1]    **Jun Li**[1]
**Xiangyu Zhang**[12] *

MEGVII Technology[1]
Beijing Academy of Artificial Intelligence[2]
`{caiyuxuan, zhouyizhuang, hanqi, zhangxiangyu}@megvii.com`

## ABSTRACT

We propose a new neural network design paradigm *Reversible Column Network (RevCol)*. The main body of RevCol is composed of multiple copies of subnetworks, named *columns* respectively, between which multi-level reversible connections are employed. Such architectural scheme attributes RevCol very different behavior from conventional networks: during forward propagation, features in RevCol are learned to be gradually *disentangled* when passing through each column, whose total information is maintained rather than *compressed* or discarded as other network does. Our experiments suggest that CNN-style RevCol models can achieve very competitive performances on multiple computer vision tasks such as image classification, object detection and semantic segmentation, especially with large parameter budget and large dataset. For example, after ImageNet-22K pre-training, RevCol-XL obtains 88.2% ImageNet-1K accuracy. Given more pre-training data, our largest model RevCol-H reaches **90.0%** on ImageNet-1K, **63.8%** $AP_{box}$ on COCO detection minival set, **61.0%** mIoU on ADE20k segmentation. To our knowledge, it is the best COCO detection and ADE20k segmentation result among *pure* (static) CNN models. Moreover, as a general macro architecture fashion, RevCol can also be introduced into transformers or other neural networks, which is demonstrated to improve the performances in both computer vision and NLP tasks. We release code and models at `https://github.com/megvii-research/RevCol`

## 1  INTRODUCTION

*Information Bottleneck principle (IB)* (Tishby et al., 2000; Tishby & Zaslavsky, 2015) rules the deep learning world. Consider a typical supervised learning network as in Fig. 1 (a): layers close to the input contain more low-level information, while features close to the output are rich in semantic meanings. In other words, information unrelated to the target is gradually *compressed* during the layer-by-layer propagation. Although such learning paradigm achieves great success in many practical applications, it might not be the optimal choice in the view of *feature learning* – down-stream tasks may suffer from inferior performances if the learned features are *over* compressed, or the learned semantic information is irrelevant to the target tasks, especially if a significant domain gap exists between the source and the target tasks (Zamir et al., 2018). Researchers have devoted great efforts to make the learned features to be more universally applicable, *e.g.* via self-supervised pre-training(Oord et al., 2018; Devlin et al., 2018; He et al., 2022; Xie et al., 2022) or multi-task learning (Ruder, 2017; Caruana, 1997; Sener & Koltun, 2018).

In this paper, we mainly focus on an alternative approach: building a network to learn *disentangled representations*. Unlike *IB* learning, disentangled feature learning (Desjardins et al., 2012; Bengio et al., 2013; Hinton, 2021) does not intend to extract the most related information while discard the less related; instead, it aims to embed the task-relevant concepts or semantic words into a few decoupled dimensions respectively. Meanwhile the whole feature vector roughly maintains as much information as the input. It is quite analogous to the mechanism in biological cells (Hinton, 2021; Lillicrap et al., 2020) – each cell shares an identical copy of the whole genome but has different

*Corresponding author. This work is supported by The National Key Research and Development Program of China (No. 2017YFA0700800) and Beijing Academy of Artificial Intelligence (BAAI).

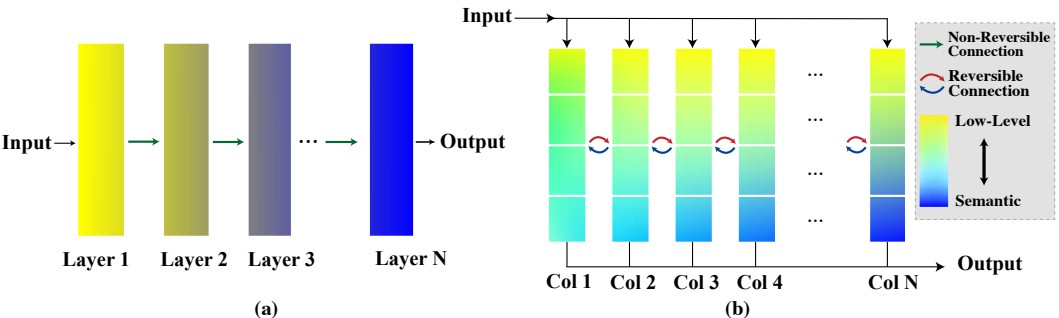

Figure 1: Sketch of the information propagation in: (a) Vanilla single-column network. (b) Our reversible column network. Yellow color denotes low-level information and blue color denotes semantic information.

expression intensities. Accordingly in computer vision tasks, learning disentangled features is also reasonable: for instance, high-level semantic representations are tuned during *ImageNet* pre-training, meanwhile the low-level information (*e.g.* locations of the edges) should also be maintained in other feature dimensions in case of the demand of down-stream tasks like object detection.

Fig. 1 (b) sketches our main idea: *Reversible Column Networks (RevCol)*, which is greatly inspired by the big picture of *GLOM* (Hinton, 2021). Our network is composed of $N$ subnetworks (named *columns*) of identical structure (however whose weights are not necessarily the same), each of which receives a copy of the input and generates a prediction. Hence multi-level embeddings, *i.e.* from low-level to highly semantic representations, are stored in each column. Moreover, *reversible transformations* are introduced to propagate the multi-level features from $i$-th column to $(i+1)$-th column without information loss. During the propagation, since the complexity and nonlinearity increases, the quality of all feature levels is expected to gradually improve. Hence the last column (Col $N$ in Fig. 1 (b)) predicts the final disentangled representations of the input.

In RevCol, one of our key contributions is the design of the *reversible transformations* between adjacent columns. The concept is borrowed from the family of *Reversible Networks* (Chang et al., 2018; Gomez et al., 2017; Jacobsen et al., 2018; Mangalam et al., 2022); however, conventional reversible structures such as *RevNets* (Gomez et al., 2017) (Fig. 2 (a)) usually have two drawbacks: first, feature maps within a reversible block are restricted to have the same shape*; second, the last two feature maps in RevNets have to contain both low-level and high-level information due to the reversible nature, which may be difficult to optimize as in conflict with *IB* principle. In this paper, we overcome the drawbacks by introducing a novel reversible multi-level fusion module. The details are discussed in Sec. 2.

We build a series of CNN-based *RevCol* models under different complexity budgets and evaluate them in mainstream computer vision tasks, such as *ImageNet* classification, *COCO* object detection and instance segmentation, as well as *ADE20K* semantic segmentation. Our models achieve comparable or better results than sophisticated CNNs or *vision transformers* like ConvNeXt (Liu et al., 2022b) and Swin (Liu et al., 2021). For example, after ImageNet-22K pre-training, our RevCol-XL model obtains **88.2%** accuracy on ImageNet-1K without using transformers or large convolutional kernels (Ding et al., 2022b; Liu et al., 2022b; Han et al., 2021). More importantly, we find RevCol can scale up well to large models and large datasets. Given a larger private pre-training dataset, our biggest model RevCol-H obtains **90.0%** accuracy on ImageNet-1K classification, **63.8%** $AP_{box}$ on COCO detection minival set, and **61.0%** mIoU on ADE20K segmentation, respectively. To our knowledge, it is the best reversible model on those tasks, as well as the best *pure* CNN model on COCO and ADE20K which only involves static kernels without dynamic convolutions (Dai et al., 2017; Ma et al., 2020). In the appendix, we further demonstrate RevCol can work with transformers (Dosovitskiy et al., 2020; Devlin et al., 2018) and get improved results on both computer vision and NLP tasks. Finally, similar to RevNets (Gomez et al., 2017), RevCol also shares the bonus of memory saving from the reversible nature, which is particularly important for large model training.

---

*In precise, feature maps of odd and even indexes should be equal sized respectively.

**Relation to previous works.** Although our initial idea on feature disentangling is derived from *GLOM* (Hinton, 2021), in *RevCol* there are a lot of simplifications and modifications. For example, GLOM suggests contrastive auxiliary loss to avoid feature collapse. Contrastive training methods need extra pairs of positive and negative samples, which is complicated and unstable. In RevCol, reversible transformations between columns provides lossless information propagation by nature. As for other multi-scale grid-like architectures such as *HRNets* (Wang et al., 2020), *DEQ models* (Bai et al., 2020) and *FPNs* (Lin et al., 2017; Tan et al., 2020), the design purpose of those models is to fuse multi-scale features rather than learn disentangled representations; therefore, in general they still follow the paradigm in Fig. 1 (a) – neither multiple entrances/exits nor reversible structures are employed. Based on those grid-like network topology, NAS based works (Ding et al., 2021; Wu et al., 2021; Liu et al., 2019; Ghiasi et al., 2019) search the optimized topology of network architectures for specific dataset. However, the RevCol architecture is not limit to specific tasks or datasets. With the reversible nature, our method maintains lossless information propagation and benefits for not only pre-training but also other down-stream tasks. Very recently, *RevBiFPN* (Chiley et al., 2022) comes up with an reversible variant of FPN, which is further employed in an HRNet-like architecture. Though our RevCol shares the similar idea of multi-scale reversible transformations with RevBiFPN, our work is done independently, which is derived from a different motivation of feature disentangling, and has much simpler architectures (*e.g.* free of reversible upsampling tower) and higher performances. We compare some of those models in Sec. 3.

## 2 METHOD

In this section, we introduce the design details of our *Reversible Column Networks (RevCol)*. Fig. 1 (b) illustrates the top-level architecture. Notice that for each column in RevCol, for simplicity we directly reuse existing structures such as *ConvNeXt* (Liu et al., 2022b), hence in the following subsections, we mainly focus on how to build the reversible connections between columns. In addition, we introduce an plug-and-play intermediate supervision on top of each column, which further improves the training convergence and feature quality.

### 2.1 MULTI-LEVEL REVERSIBLE UNIT

In our network, *reversible transformations* plays a key role in feature disentangling without information loss, whose insight comes from *Reversible Neural Networks* (Dinh et al., 2014; Chang et al., 2018; Gomez et al., 2017; Jacobsen et al., 2018; Mangalam et al., 2022). Among them, we first take a review of one representative work *RevNet* (Gomez et al., 2017). As shown in Fig. 2 (a), RevNet first partitions the input $x$ into two groups, $x_0$ and $x_1$. Then for later blocks, for example, block $t$, it takes two anterior blocks' outputs $x_{t-1}$ and $x_{t-2}$ as input and generates the output $x_t$. The mapping of block $t$ is *reversible*, i.e. $x_{t-2}$ can be reconstructed by two posterior blocks $x_{t-1}$ and $x_t$. Formally, the forward and *inverse* computation follow the equations [†]:

$$\begin{aligned} Forward : x_t &= \boldsymbol{F}_t(x_{t-1}) + \gamma x_{t-2} \\ Inverse : x_{t-2} &= \gamma^{-1}[x_t - \boldsymbol{F}_t(x_{t-1})], \end{aligned} \quad (1)$$

where $\boldsymbol{F}_t$ denotes an arbitrary non-linear operation analogous to those residual functions in standard *ResNets*; $\gamma$ is a simple reversible operation (*e.g.* channel-wise scaling), whose inverse is denoted by $\gamma^{-1}$. As mentioned in the introduction, the above formulation involves too strong constraint on the feature dimensions, i.e. $x_t, x_{t+2}, x_{t+4}, ...$ have to be equal sized, which is not flexible in architecture design. That is why RevNets (Gomez et al., 2017) introduce some non-reversible down-sampling blocks between reversible units, hence the whole network is not fully reversible. More importantly, we find there is no clear way to directly employ Eq. 1 to bridge the columns in Fig. 1 (b).

To address the issue, we generalize Eq. 1 into the following form:

$$\begin{aligned} Forward : x_t &= \boldsymbol{F}_t(x_{t-1}, x_{t-2}, ..., x_{t-m+1}) + \gamma x_{t-m} \\ Inverse : x_{t-m} &= \gamma^{-1}[x_t - \boldsymbol{F}_t(x_{t-1}, x_{t-2}, ..., x_{t-m+1})], \end{aligned} \quad (2)$$

---

[†]In Gomez et al. (2017), the proposed reversible equations are formulated as $y_1 = x_1 + \mathcal{F}(x_2)$ and $y_2 = x_2 + \mathcal{G}(y_1)$. While in this paper, we reformulate those notations $y_2, y_1, x_2, x_1, \mathcal{G}, \mathcal{F}$ as $x_t, x_{t-1}, x_{t-2}, x_{t-3}, \boldsymbol{F}_t, \boldsymbol{F}_{t-1}$, respectively, in order to better illustrate the relation between building block $t$ and $t-1$. It is easy to prove the two formulations are equivalent.

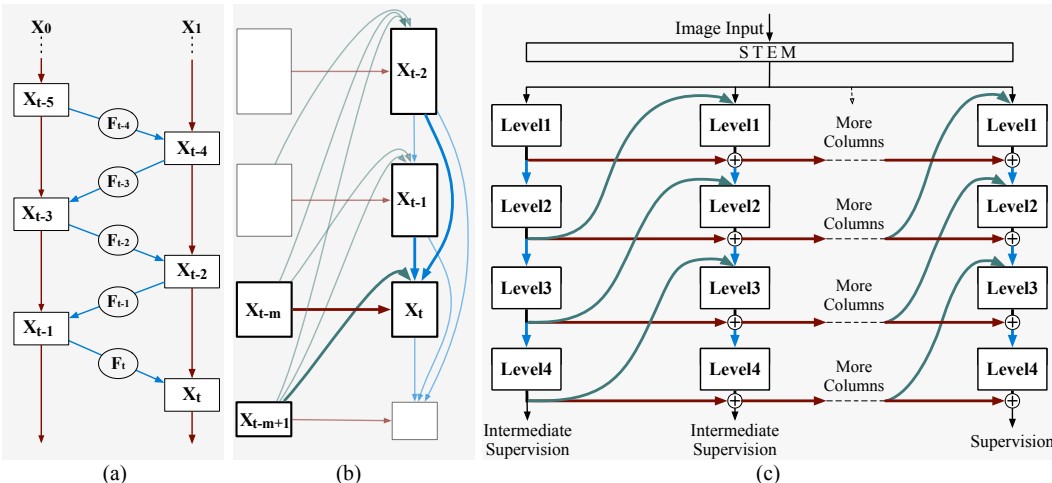

Figure 2: (a) Reversible unit in RevNet (Gomez et al., 2017). (b) Multi-level reversible unit. All inputs for level $t$ are highlighted. (c) An overview of the whole reversible column network architecture, with simplified multi-level reversible unit.

where $m$ is the *order* of the recursion ($m \geq 2$). Clearly, the extension is still reversible. Then we partition every $m$ feature maps into a group: $(x_1, x_2, \ldots, x_m), (x_{m+1}, x_{m+2}, \ldots, x_{2m}), \ldots$. Given the features within any of the group, we can easily compute the features in other groups recursively according to Eq. 2. Compared with the original form, Eq. 2 has the following two nice properties:

- The constraint on the feature map sizes is greatly relaxed if $m$ is relatively large. Notice that Eq. 1 does not require feature maps *within* each group to be equal sized; such constraint only exist between groups. Therefore, we can use tensors of different shape to represent features of different semantic levels or different resolutions.

- Eq. 2 can easily cooperate with existing network architectures, even though the latter is not reversible. For example, we can assign $m$ feature maps in a standard *ResNet* to represent the feature maps within a group $(x_t, x_{t+1}, \ldots, x_{t+m-1})$, which is still compatible with Eq. 2 since ResNet can be viewed as a part of $(\boldsymbol{F}_t, \boldsymbol{F}_{t+1}, \ldots, \boldsymbol{F}_{t+m-1})$ respectively. Thus the whole network is still reversible.

Therefore, we can reorganize Eq. 2 into a *multi-column* fashion, as shown in Fig. 2 (b). Each column is composed of $m$ feature maps within a group, as well as their mother network. We name it *multi-level reversible unit*, which is the basic component of our RevCol as in Fig. 1 (b).

## 2.2 REVERSIBLE COLUMN ARCHITECTURE

### 2.2.1 MACRO DESIGN

As discussed in the introduction (see Fig. 1 (b)), our network *RevCol* is composed of multiple subnetworks with *reversible connections* to perform feature disentangling. Fig. 2 (c) elaborates the architecture design. Following the common practice of recent models (Dosovitskiy et al., 2020; Liu et al., 2022b), first the input image is split into non-overlapping patches by a patch embedding module. After that, patches are fed into each subnetwork (column). Columns can be implemented with any conventional single-column architectures, *e.g. ViT* (Dosovitskiy et al., 2020) or *ConvNeXt* (Liu et al., 2022b). We extract four-level feature maps from each column to propagate information between columns; for example, if the columns are implemented with widely-used hierarchical networks (Liu et al., 2021; He et al., 2016; Liu et al., 2022b), we can simply extract multi-resolution features from the output of each stage. For classification tasks, we only use feature map of the last level (Level 4) in the last column for rich semantic information. For other down-stream tasks like object detection and semantic segmentation, we use feature maps of all the four levels in the last column as they contain both low-level and semantic information.

To implement the reversible connections between columns, we adopt the *multi-level reversible unit* proposed in Eq. 2, but in a simplified fashion: rather than take $(m-1)$ inputs for each non-linear operation $\boldsymbol{F}_t(\cdot)$, we use only one low-level feature $x_{t-1}$ at the current column and one high-level feature $x_{t-m+1}$ at the previous column as the input. The simplification does not break the reversible property. We find more inputs bring minor accuracy gain but consume much more GPU resources. Thus Eq. 2 is simplified as:

$$
\begin{aligned}
Forward &: x_t = \boldsymbol{F}_t(x_{t-1}, x_{t-m+1}) + \gamma x_{t-m} \\
Inverse &: x_{t-m} = \gamma^{-1}[x_t - \boldsymbol{F}_t(x_{t-1}, x_{t-m+1})].
\end{aligned}
\tag{3}
$$

Compared with conventional architectures, the *macro design* of our RevCol has the following three properties or advantages:

**Feature disentangling.** In RevCol, the lowest level of each column maintains low-level features as it is close to the input, while the highest level in the last column is highly semantic because it is directly connected to the supervision. Therefore, information in different levels is gradually *disentangled* during the (lossless) propagation between columns – some feature maps are more and more semantic and some maintain to be low-level. Detailed analyses are presented in Appendix G. The property brings many potential advantages, for instance, more flexible to downstream tasks which rely on both high-level and low-level features. We argue that *reversible connection* plays a key role in the disentangling mechanism – some previous works like *HRNet* (Wang et al., 2020) involve multi-level feature fusion but without reversible connection, which may suffer from information loss and lead to inferior performances in our experiments (see Section D.2).

**Memory saving.** The training of conventional networks takes a lot of memory footprint to store the activations during forward propagation as the demand of gradient computation. While in our RevCol, since the connections between columns are *explicitly* reversible, during the back-propagation we can reconstruct the required activations *on the fly* from the last column to the first, which means we only need to maintain activations from *one* column in memory during training. In Section D.4, we demonstrate RevCol costs roughly $\mathcal{O}(1)$ additional memory with the increase of column numbers.

**New scaling factor for big models.** In RevCol architecture, column number serves as a new dimension in addition to depth (number of blocks), and width (channels of each block) in vanilla single-column CNNs or ViTs. Increasing column numbers has similar income as increasing both width and depth in certain range.

### 2.2.2 MICRO DESIGN

We employ *ConvNeXt* blocks (Liu et al., 2022b) to implement each column in our network by default; other architectures, such as *transformers*, are also applicable (see Appendix B for details). We make a few modifications to make ConvNeXt compatible with our macro architecture:

**Fusion module.** As shown in Fig. 3, in each level of original ConvNeXt, the inputs are first down-sampled in a patch-merging block. Then the outputs are passed through a bunch of residual blocks. In RevCol, we introduce a fusion module to fuse the feature maps from the current and previous columns (refer to Fig. 2 (c), green and blue connections). We modify the original patch-merging block in ConvNeXt by putting the LayerNorm after the patch-merging convolution rather than before. Channel numbers are doubled in patch-merging convolution. We also introduce an up-sampling block, which is composed of a linear channel mapping layer, a LayerNorm normalization and a feature map interpolation layer. We halve the channel numbers in linear channel mapping layer. The outputs of the two blocks are summed up and then passed to the residual blocks followed by.

**Kernel size.** In RevCol we revise the $7 \times 7$ convolutions in original ConvNeXt (Liu et al., 2022b) to $3 \times 3$ by default, mainly to speed up the training. Increasing kernel size further obtains more accuracy, but not very much, partly because the our multi-column design enlarges the effective receptive field. Please refer to Section D.5 for more details.

**Reversible operation $\gamma$.**  We adopt a learnable reversible channel-wise scaling as reversible operation $\gamma$ to keep the network stable. Each time the features are summed up in forward of Eq. 3, the magnitude grows larger, which makes the training process unstable. Using a learnable scaling can suppress the magnitude of features. During training, we truncate the absolute value of $\gamma$ so that it will never be smaller than $1e^{-3}$, because the numerical error could become large in the reverse computation when $\gamma$ is too small.

## 2.3 INTERMEDIATE SUPERVISION

Though *multi-level reversible unit* is able to maintain information during column iteration, the down-sample block still can discard information *inside* column. Features at the end of the front columns is too close to the final output, for reversible connections simply do scaling and summation. Such information loss leads to inferior performance. Similar problem also happens when using deeply-supervised method (Lee et al., 2015; Szegedy et al., 2015).

To mitigate the problem of information collapse, we propose an intermediate supervision method which adds additional supervision into front columns. For features in front columns, we hope to keep the mutual information between features and the input image as much as possible, so that the network discard less information within columns. Consider *RevCol* gradually disentangle semantic and low-level information, extracting and leveraging the task-relevant information can further boost the performance. Therefore, we need to maximize the lower bound of mutual information between features and the prediction.

Inspired by Wang et al. (2021), we add two auxiliary heads to last level features (Level 4). One is a decoder (He et al., 2022) which reconstructs the input image, the other is a linear classifier. The linear classifier can be trained in a regular classification fashion with the *cross-entropy (CE)* loss. The parameters of decoder are optimized by minimizing the *binary cross-entropy (BCE)* reconstruction loss. Compared with commonly used *L1* and *L2* loss, interpreting the distribution of reconstructed logits and input image as *bit probabilities (Bernoullis)* outputs smoother value, which makes it more compatible with CE Loss.

For intermediate supervision at one column, the compound loss is the weighted summation of the above two loss. Note that supervision heads may not be added to all columns. For all the variants of RevCol, we set the number of compound loss to 4 empirically (eg. for a 8 column RevCol, the supervision heads are added to column 2, 4, and 6, and 8).

The total loss $L$ in is the summation of all compound loss:

$$L = \sum_{i=1}^{n} (\alpha_i \mathcal{L}_{\mathrm{BCE}} + \beta_i \mathcal{L}_{\mathrm{CE}}) \tag{4}$$

$n$ denotes the total number of compound loss. $\mathcal{L}_{\mathrm{BCE}}$ and $\mathcal{L}_{\mathrm{CE}}$ denotes BCE loss and CE loss correspondingly. $\alpha_i$ and $\beta_i$ are changed linearly with the compound loss number. When the compound loss is added in earlier columns, we use larger value $\alpha_i$ and smaller value $\beta_i$ to keep $I(\boldsymbol{h}, x)$. In later columns, value $\alpha_i$ decreases and $\beta_i$ increases, which helps boost the performance.

## 3 EXPERIMENTS

We construct different *RevCol* variants, RevCol-T/S/B/L, to be of similar complexities to Swin transformers and ConvNeXts. We also build a larger RevCol-XL and RevCol-H to test the scaling up capability. These variants adopt different channel dimension $C$, blocks in each column $B$ and column numbers *COL*. The configuration hyper-parameters of these model variants are:

- RevCol-T:  $C = (64, 128, 256, 512)$,  $B = (2, 2, 4, 2)$,  $COL = 4$
- RevCol-S:  $C = (64, 128, 256, 512)$,  $B = (2, 2, 4, 2)$,  $COL = 8$
- RevCol-B:  $C = (72, 144, 288, 576)$,  $B = (1, 1, 3, 2)$,  $COL = 16$
- RevCol-L:  $C = (128, 256, 512, 1024)$,  $B = (1, 2, 6, 2)$,  $COL = 8$
- RevCol-XL:  $C = (224, 448, 896, 1792)$,  $B = (1, 2, 6, 2)$,  $COL = 8$
- RevCol-H:  $C = (360, 720, 1440, 2880)$,  $B = (1, 2, 6, 2)$,  $COL = 8$

Table 1: **ImageNet classification results.** We compare our models with state-of-the-art ○ Vision Transformers and ● CNNs that have comparable FLOPs and parameters. ↑ denotes models fine-tuning using image size larger than $224^2$. We report the top-1 accuracy on the validation set of ImageNet as well as the number of parameters and FLOPs. Our models are highlighted in gray.

| Model | Image Size | Params (M) | FLOPs (G) | Top-1 Acc. |
|---|---|---|---|---|
| *ImageNet-1K trained models* | | | | |
| ○ Swin-T (Liu et al.) | $224^2$ | 28 | 4.5 | 81.3 |
| ○ DeiT-S (Touvron et al. | $224^2$ | 22 | 4.6 | 79.8 |
| ○ Rev-ViT-S (Mangalam et al.) | $224^2$ | 22 | 4.6 | 79.9 |
| ● RevBiFPN-S3 (Chiley et al.) | $288^2$ | 20 | 3.3 | 81.1 |
| ● EfficientNet-B4 (Tan & Le) | $380^2$ | 19 | 4.2 | 82.9 |
| ● ConvNeXt-T (Liu et al.) | $224^2$ | 29 | 4.5 | 82.1 |
| ● RevCol-T | $224^2$ | 30 | 4.5 | 82.2 |
| ○ Swin-S (Liu et al.) | $224^2$ | 50 | 8.7 | 83.0 |
| ○ MViTv1-B (Fan et al.) | $224^2$ | 37 | 7.8 | 83.0 |
| ○ T2T-ViT-19 (Yuan et al.) | $224^2$ | 39 | 8.4 | 81.4 |
| ● RevBiFPN-S4 (Chiley et al.) | $320^2$ | 49 | 10.6 | 83.0 |
| ● EfficientNet-B5  Tan & Le) | $456^2$ | 30 | 9.9 | 83.6 |
| ● ConvNeXt-S (Liu et al.) | $224^2$ | 50 | 8.7 | 83.1 |
| ● RevCol-S | $224^2$ | 60 | 9.0 | 83.5 |
| ○ Swin-B (Liu et al.) | $224^2$ | 89 | 15.4 | 83.5 |
| ○ DeiT-B (Touvron et al.) | $224^2$ | 86 | 17.5 | 81.8 |
| ○ Rev-ViT-B(Mangalam et al.) | $224^2$ | 87 | 17.6 | 81.8 |
| ● RepLKNet-31B (Ding et al.) | $224^2$ | 79 | 15.3 | 83.5 |
| ● RevBiFPN-S5 (Chiley et al.) | $352^2$ | 82 | 21.8 | 83.7 |
| ● EfficientNet-B6 (Tan & Le) | $528^2$ | 43 | 19.0 | 84.0 |
| ● ConvNeXt-B (Liu et al.) | $224^2$ | 88 | 15.4 | 83.8 |
| ● RevCol-B | $224^2$ | 138 | 16.6 | 84.1 |

| Model | Image Size | Params (M) | FLOPs (G) | Top-1 Acc. |
|---|---|---|---|---|
| *ImageNet-22K pre-trained models (ImageNet-1K fine-tuned)* | | | | |
| ○ Swin-B (Liu et al. | $224^2$ | 88 | 15.4 | 85.2 |
| ○ Swin-B↑ (Liu et al. | $384^2$ | 88 | 47.0 | 86.4 |
| ○ ViT-B↑ (Dosovitskiy et al.) | $384^2$ | 86 | 55.4 | 84.0 |
| ● RepLKNet-31B (Ding et al.) | $224^2$ | 79 | 15.3 | 85.2 |
| ● RepLKNet-31B↑ (Ding et al.) | $384^2$ | 79 | 45.1 | 86.0 |
| ● ConvNeXt-B (Liu et al.) | $224^2$ | 89 | 15.4 | 85.8 |
| ● ConvNeXt-B↑ (Liu et al.) | $384^2$ | 89 | 45.1 | 86.8 |
| ● RevCol-B | $224^2$ | 138 | 16.6 | 85.6 |
| ● RevCol-B↑ | $384^2$ | 138 | 48.9 | 86.7 |
| ○ Swin-L (Liu et al.) | $224^2$ | 197 | 34.5 | 86.3 |
| ○ Swin-L↑ (Liu et al.) | $384^2$ | 197 | 103.9 | 87.3 |
| ○ ViT-L↑ (Dosovitskiy et al.) | $384^2$ | 307 | 190.7 | 85.2 |
| ● RepLKNet-31L (Ding et al.) | $384^2$ | 172 | 96.0 | 86.6 |
| ● ConvNeXt-L (Liu et al.) | $224^2$ | 198 | 34.4 | 86.6 |
| ● ConvNeXt-L↑ (Liu et al.) | $384^2$ | 198 | 101.0 | 87.5 |
| ● RevCol-L | $224^2$ | 273 | 39.0 | 86.6 |
| ● RevCol-L↑ | $384^2$ | 273 | 116.0 | 87.6 |
| ● ConvNeXt-XL↑  (Liu et al.) | $384^2$ | 350 | 179.0 | 87.8 |
| ● RevCol-XL↑ | $384^2$ | 834 | 350.0 | 88.2 |
| *Extra data pre-trained models (ImageNet-1K fine-tuned)* | | | | |
| ● RevCol-XL↑ | $384^2$ | 834 | 350.0 | 89.4 |
| ● RevCol-H↑ | $640^2$ | 2158 | 2537 | 90.0 |

We conduct image classification on *ImageNet* dataset (Deng et al., 2009; Ridnik et al., 2021). We also test our models on the downstream object detection task and semantic segmentation task on commonly used *MS-COCO* (Lin et al., 2014) and *ADE20k* (Zhou et al., 2017b) dataset. Training and fine-tuning settings please refer to Appendix F. Furthermore, we show the performance of RevCol with transformer on vision and language tasks (shown in Appendix B).

## 3.1    IMAGE CLASSIFICATION

On ImageNet (1.28M images) (Deng et al., 2009) dataset, we train RevCol  for 300 epochs with intermediate supervision. Hyperparameters, augmentation and regularization strategies follows Liu et al. (2022b) We also pre-train our models on the larger ImageNet-22K dataset (Ridnik et al., 2021), which contains 14.2 million images.

In Tab. 1, we compare our RevCol  variants with commonly used recent *Transformers* and *CNNs* on ImageNet-1k validation set. Our models outperforms a large number of vanilla single-column CNNs and Transformers with similar complexities. For example, RevCol-S achieve 83.5% Top-1 accuracy, outperform ConvNeXt-S by 0.4 points. When pre-trained with larger ImageNet-22K dataset, RevCol-XL achieves 88.2% Top-1 accuracy. As RevCol maintains some task-irrelevant low-level information in classification pre-training, relaxing the constraint of params and FLOPs and enlarging dataset size can further boost our models' performance. To further test the scaling up effectiveness of large dataset, we build a 168-million-image semi-labeled dataset (see Appendix E). With extra data pre-training and ImageNet-1k fine-tuning, our RevCol-H achieves **90.0% top-1 accuracy**. Our results further demonstrate with RevCol, CNN models can also share the dividends of large model and massive data pre-training.

## 3.2    OBJECT DETECTION

We evaluate our proposed RevCol on object detection task.  Experiments are conducted on the MS-COCO dataset using the Cascade Mask R-CNN (Cai & Vasconcelos, 2019) framework. We

Table 2: **Object detection results on MS-COCO dataset with different backbones.** We report box AP and mask AP with single scale testing on COCO minival set. FLOPs are measured under input sizes of (1280, 800).

| Backbone | $AP^{box}$ | $AP^{box}_{50}$ | $AP^{box}_{75}$ | $AP^{mask}$ | $AP^{mask}_{50}$ | $AP^{mask}_{75}$ | Params | FLOPs |
|---|---|---|---|---|---|---|---|---|
| *ImageNet-1K pre-trained* | | | | | | | | |
| ○ Swin-T (Liu et al.) | 50.5 | 69.3 | 54.9 | 43.7 | 66.6 | 47.1 | 86M | 745G |
| ● ConvNeXt-T (Liu et al.) | 50.4 | 69.1 | 54.8 | 43.7 | 66.5 | 47.3 | 86M | 741G |
| ● RevCol-T | 50.6 | 68.9 | 54.9 | 43.8 | 66.7 | 47.4 | 88M | 741G |
| ○ Swin-S (Liu et al.) | 51.8 | 70.4 | 56.3 | 44.7 | 67.9 | 48.5 | 107M | 838G |
| ● ConvNeXt-S (Liu et al.) | 51.9 | 70.8 | 56.5 | 45.0 | 68.4 | 49.1 | 108M | 827G |
| ● RevCol-S | 52.6 | 71.1 | 56.8 | 45.5 | 68.8 | 49.0 | 118M | 833G |
| ○ Swin-B (Liu et al.) | 51.9 | 70.9 | 56.5 | 45.0 | 68.4 | 48.7 | 145M | 982G |
| ● ConvNeXt-B (Liu et al.) | 52.7 | 71.3 | 57.2 | 45.6 | 68.9 | 49.5 | 146M | 964G |
| ● RepLKNet-B (Ding et al.) | 52.2 | - | - | 45.2 | - | - | 137M | 965G |
| ● RevCol-B | 53.0 | 71.4 | 57.3 | 45.9 | 69.1 | 50.1 | 196M | 988G |
| *ImageNet-22K pre-trained* | | | | | | | | |
| ○ Swin-B (Liu et al.) | 53.0 | 71.8 | 57.5 | 45.8 | 69.4 | 49.7 | 145M | 982G |
| ● ConvNeXt-B (Liu et al.) | 54.0 | 73.1 | 58.8 | 46.9 | 70.6 | 51.3 | 146M | 964G |
| ● RepLKNet-B (Ding et al.) | 53.0 | - | - | 46.3 | - | - | 137M | 965G |
| ● RevCol-B | 55.0 | 73.5 | 59.7 | 47.5 | 71.1 | 51.8 | 196M | 988G |
| ○ Swin-L (Liu et al.) | 53.9 | 72.4 | 58.8 | 46.7 | 70.1 | 50.8 | 253M | 1382G |
| ● ConvNeXt-L (Liu et al.) | 54.8 | 73.8 | 59.8 | 47.6 | 71.3 | 51.7 | 255M | 1354G |
| ● RepLKNet-L (Ding et al.) | 53.9 | - | - | 46.5 | - | - | 229M | 1321G |
| ● RevCol-L | 55.9 | 74.1 | 60.7 | 48.4 | 71.8 | 52.8 | 330M | 1453G |
| *Extra data pre-trained* | | | | | | | | |
| ● RevCol-H (HTC++) | 61.1 | 78.8 | 67.0 | 53.0 | 76.3 | 58.7 | 2.41G | 4417G |
| ● RevCol-H (Objects365+DINO) | 63.8 | 81.8 | 70.2 | - | - | - | 2.18G | 4012G |

also finetune our largest model RevCol-H with HTC++ (Chen et al., 2019) and DINO (Zhang et al., 2022a) Framework.

In Tab. 2, we compare the $AP_{box}$ and $AP_{mask}$ with Swin/ConvNeXt in variant sizes on COCO validation set. We find RevCol models surpass other counterparts with similar computation complexities. Information retained in pre-training helps RevCol models acheieve better results in down-stream tasks. When the model size grows larger, this advantage becomes more remarkable. After finetuning under Objects365(Shao et al., 2019) dataset and DINO (Zhang et al., 2022a) framework, our largest model RevCol-H achieves **63.8%** $AP_{box}$ on COCO detection minival set.

## 3.3 SEMANTIC SEGMENTATION

We also evaluate RevCol backbones on the ADE20K semantic segmentation task with *UperNet* (Xiao et al., 2018) framework. We do not use intermediate-supervision in down-stream fine-tune process. To further explore our model's capacity and reach the leading performance, we utilize recent segmentation framework *Mask2Former* (Cheng et al., 2022), and adopt the same training settings.

In Tab. 3, we report validation *mIoU* with single-scale and multi-scale flip testing. RevCol models can achieve competitive performance across different model capacities, further validating the effectiveness of our architecture design. It's worth noting that when use Mask2Former detector and extra pre-training data, RevCol-H achieves an mIoU of 61.0%, which shows feasible scalability towards large-scale vision applications.

## 4 RELATED WORKS

### 4.1 DISENTANGLE REPRESENTATION LEARNING AND PART-WHOLE HIERARCHY

A disentangled representation is generally described as one which separates the factors of variation, explicitly representing the important attributes of the data (Desjardins et al., 2012; Bengio et al., 2013). Desjardins et al. (2012); Kulkarni et al. (2015); Higgins et al. (2017); Chen et al. (2016); Karras et al. (2019) seek to learn disentangled representations through generative models. Locatello et al. (2019) points out that unsupervised learning of disentangled representations is fundamentally impossible without inductive biases both on the considered learning approaches and the datasets. The recent proposal of *GLOM* (Hinton, 2021) gives an idea of representing a part-whole hierarchy by a

Table 3: **Semantic segmentation result on ADE20k dataset with different backbones.** we report mIoU results with single/multi-scale testing. FLOPs are measured under input sizes of (2048, 512), (2560, 640) for IN-1K and IN-22K pre-trained models respectively.

| Backbone | crop size | $mIoU_{ss}$ | $mIoU_{ms}$ | Params | FLOPs |
|---|---|---|---|---|---|
| *ImageNet-1K pre-trained* | | | | | |
| ○ Swin-T (Liu et al.) | $512^2$ | 44.5 | 45.8 | 60M | 945G |
| ● ConvNeXt-T (Liu et al.) | $512^2$ | 46.0 | 46.7 | 60M | 939G |
| ● RevCol-T | $512^2$ | 47.4 | 47.6 | 60M | 937G |
| ○ Swin-S (Liu et al.) | $512^2$ | 47.6 | 49.5 | 81M | 1038G |
| ● ConvNeXt-S (Liu et al.) | $512^2$ | 48.7 | 49.6 | 82M | 1027G |
| ● RevCol-S | $512^2$ | 47.9 | 49.0 | 90M | 1031G |
| ○ Swin-B (Liu et al.) | $512^2$ | 48.1 | 49.7 | 121M | 1188G |
| ● RepLKNet-B (Ding et al.) | $512^2$ | 49.9 | 50.6 | 112M | 1170G |
| ● ConvNeXt-B (Liu et al.) | $512^2$ | 49.1 | 49.9 | 122M | 1170G |
| ● RevCol-B | $512^2$ | 49.0 | 50.1 | 122M | 1169G |
| *ImageNet-22K pre-trained* | | | | | |
| ○ Swin-B (Liu et al.) | $640^2$ | 50.3 | 51.7 | 121M | 1841G |
| ● RepLKNet-B (Ding et al.) | $640^2$ | 51.5 | 52.3 | 112M | 1829G |
| ● ConvNeXt-B (Liu et al.) | $640^2$ | 52.6 | 53.1 | 122M | 1828G |
| ● RevCol-B | $640^2$ | 52.7 | 53.3 | 122M | 1827G |
| ○ Swin-L (Liu et al.) | $640^2$ | 52.1 | 53.5 | 234M | 2468G |
| ● RepLKNet-L (Ding et al.) | $640^2$ | 52.4 | 52.7 | 207M | 2404G |
| ● ConvNeXt-L (Liu et al.) | $640^2$ | 53.2 | 53.7 | 235M | 2458G |
| ● RevCol-L | $640^2$ | 53.4 | 53.7 | 306M | 2610G |
| *Extra data pre-trained* | | | | | |
| ● RevCol-H | $640^2$ | 57.8 | 58.0 | 2421M | - |
| ● RevCol-H + Mask2Former | $640^2$ | 60.4 | 61.0 | 2439M | - |

weight-sharing columns. The GLOM architecture provides an interpretable part-whole hierarchies for deep neural network (Garau et al., 2022). In RevCol, we adopt the design of using columns, but not modeling the process of formulating islands. On the contrary, our column iteration process maintains both low-level and high-level information and gradually disentangle them. Rather than using self-supervised methods, RevCol can be trained with supervision end-to-end.

## 4.2 REVERSIBLE NETWORKS

Gomez et al. (2017) firstly propose *RevNet* that allow back propagation without saving intermediate activations. The reversible design remarkably saves the training cost, since it keep $\mathcal{O}(1)$ GPU memory consumption as model depth scaling up. Jacobsen et al. (2018) propose a fully reversible network that can reverse back to the input without any information loss. Chang et al. (2018) develop a theoretical framework on stability and reversibility of deep neural network and derive reversible networks that can go arbitrarily deep. Mangalam et al. (2022) expand the reversible network scope from CNNs to Transformers. *RevBiFPN* (Chiley et al., 2022), a concurrent work of ours, add the reversible connections to *BiFPN* (Tan et al., 2020) network. Our RevCol maintains the information without loss inside each column rather than the whole BiFPN network in RevBiFPN.

## 5 CONCLUSION

In this paper, we propose RevCol, a reversible column based foundation model design paradigm. During the lossless propagation through columns, features in RevCol are learned to be gradually disentangled and the total information is still maintained rather than compressed. Our experiments suggests that RevCol can achieve competitive performance in multiple computer vision tasks. We hope RevCol could contribute to better performance in various tasks in both vision and language domains.

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

# A  MICRO DESIGN DETAILS

Figure 3: (a) Levels in ConvNeXt. Level $l$ contains a patch merging down-sample block and $n_l$ residual blocks. (b) Levels in RevCol. Level $l$ is composed of a fusion module, $n_l$ residual blocks and a reversible operation. Note that Level $l$ takes features maps $x_{t-1}$, $x_{t-m+1}$ and $x_{t-m}$ as input. Feature maps $x_{t-1}$ and $x_{t-m+1}$ are fed into the fusion module and feature maps $x_{t-m}$ are fed into the reversible operation. (c) Design of the fusion module.

In this section, we provide the architecture design details for RevCol. As depicted in Fig. 2 and Section 2.2, our RevCol contains multiple columns with reversible connections. Fig. 3 (a) shows the architecture of ConvNeXt. Note that we replace the $7 \times 7$ depth-wise convolution in ConvNeXt with $3 \times 3$, as described in Sec. 2.2.2. In Fig. 3 (b), we show in detail how to extend to our RevCol on the basis of ConvNeXt. First, we replace the down-sample block with a fusion block to fuse low-level representations in current column and high-level ones from the previous column, and Fig. 3 (c) shows the details of fusion block which contains up-sample and down-sample operations to handle different resolutions. Second, for each level, same-level representations from the previous column are added to current level's output and are ready to propagate as a whole. Thanks to the two modifications, feature maps from different hierarchies aggregate together to form the intermediate representation. In Fig. 3 (c), we use a `Linear-LayerNorm` followed by a nearest interpolation to up-sample low resolution features. A $2 \times 2$ kernel `Conv2d` with stride 2 down-samples the high resolution features, followed by a `LayerNorm` to balance the contributions of the two inputs.

# B  GENERALIZATION TO TRANSFORMERS

## B.1  VISION TRANSFORMER MODELS

RevCol contains multiple light-weight sub-networks with reversible connections. In this paper, we adopt the ConvNext micro design by default except for multi-columns fusion and smaller convolution kernel as described in Sec. 2.2.2. However, the micro design of our RevCol is not limited to convolutional networks, but is also compatible with isotropic designing, such as the vanilla vision

transformer (ViT) (Dosovitskiy et al., 2020). In this section, we show the micro design of RevCol can generalized to vanilla ViT, namely RevCol-ViT, with promising experimental results.

net-ViT maintains the feature resolution in the reversible columns. Thus the patch merging blocks and up-sample blocks in the fusion modules are replaced with a simple linear projection with a post LayerNorm. We use the vanilla ViT building block instead of the ConvNext building block variant. The post LayerNorms and normalized dot-product attention are used in ViT blocks to stabilize training convergence, similar to Liu et al. (2022a). With the properties of isotropy, we evenly arrange the building blocks in each column. The configuration details of RevCol-ViT are:

- RevCol-ViT-S: $C = (224, 224, 224, 224)$, $B = (2, 2, 2, 2)$, $HEAD = 4$, $COL = 4$
- RevCol-ViT-B: $C = (384, 384, 384, 384)$, $B = (3, 3, 3, 3)$, $HEAD = 6$, $COL = 4$

Table 4: ImageNet-1K classification results. We compare our RevCol-ViT with state-of-the-art isotropic ∘ Vision Transformers and • CNNs that have comparable FLOPs and parameters.

| Model | Image Size | Params | FLOPs | Top-1 Acc. |
|---|---|---|---|---|
| ∘ DeiT-S (Touvron et al., 2020) | $224^2$ | 22M | 4.6G | 79.8 |
| • ConvNext-S (*iso.*) (Liu et al., 2022b) | $224^2$ | 22M | 4.3G | 79.7 |
| ∘ **RevCol-ViT-S** | $224^2$ | 16M | 4.6G | **80.6** |
| ∘ ViT-B (Dosovitskiy et al., 2020) | $384^2$ | 86M | 55.4G | 77.9 |
| ∘ DeiT-B (Touvron et al., 2020) | $224^2$ | 86M | 17.6G | 81.7 |
| ∘ Rev-ViT-B (Mangalam et al., 2022) | $224^2$ | 87M | 17.6G | 81.8 |
| ∘ Rev-MViT-B (Mangalam et al., 2022) | $224^2$ | 39M | 8.7G | 82.5 |
| • ConvNext-B (*iso.*) (Liu et al., 2022b) | $224^2$ | 87M | 16.9G | 82.0 |
| ∘ **RevCol-ViT-B** | $224^2$ | 67M | 18.8G | **82.7** |

We use the same training setting with the anisotropic RevCol as described in Sec. 3.1, except that the intermediate supervision is discarded for simplicity and the stochastic depth rate is set as 0.2 for RevCol-B. We scale down the value of last linear projection layers in each FFN accroding to the network depth in initialization, same as BEiT (Bao et al., 2021). In Tab. 4, we compare the RevCol-ViT with vanilla ViT and other concurrent isotropic designs. Our RevCol-ViT surpasses vanilla vision transformer (77.9% for ViT and 81.7% for DeiT) and convolutional network ConvNeXt (82.0%) that have comparable model parameters and computational overhead on ImageNet-1k classification w.r.t. the top-1 accuracy.

## B.2 LANGUAGE MODELS

Considering the great success of applying transformer to computer vision, i.e., ViT (Dosovitskiy et al., 2020), we also made some exploration to generalize RevCol to natural language processing (NLP). Based on the design in Appendix B.1, we can easily apply the isotropic RevCol to language models with minor modification. To be specific, we replace the stem module in our RevCol with word embedding and positional encoding in transformer. Then, the RevCol can be plugged into the original transformer as an encoder. The output of the last column in RevCol will be used as the memory keys and values for the attention layers in decoder, just exactly the same as the original transformer.

We select the translation task to evaluate the potential of the RevCol in NLP. We run experiments on the WMT'16 English-German (En-De) dataset with 4.5M sentences and larger WMT'14 English-French dataset with 36M sentences. Each sentence is encoded by joint source and target byte pair encoding following Sennrich et al. (2016). The details of model architecture and the BLEU score are shown in Tab. 5.

All the dataset preparation and the training configurations follows Ott et al. (2018) and the open source project `fairseq`. The models were trained for 300K steps with batch-size of 28,672 tokens on En-De and 200K steps with batch-size of 86,016 on En-Fr. We discard the intermediate supervision for simplicity. As shown in Tab. 5, our RevCol outperforms vanilla transformer with comparable parameters on En-De (28.67 *vs*. 28.43) and En-Fr (43.40 *vs*. 43.07), which demonstrates the RevCol's applicability to NLP.

Table 5: BLEU score on newstest2014 for WMT English-German (En-De) and English-French (En-Fr) translation task. $^\dagger$ indicates we re-run the experiments with `fairseq`.

| Model | Encoder | | | | Decoder | | | | Params | Task | BLEU |
|---|---|---|---|---|---|---|---|---|---|---|---|
| | arch | $d_{model}$ | $d_{ff}$ | head | arch | $d_{model}$ | $d_{ff}$ | head | | | |
| Transformer$^\dagger_{big}$ (Vaswani et al., 2017) | $N = 6$ | 1024 | 4096 | 16 | $N = 6$ | 1024 | 4096 | 16 | 209M 221M | En-De En-Fr | 28.43 43.07 |
| **RevCol-Transformer** | $B = (1,1,1,1)$ $COL = 4$ | 768 | 3072 | 12 | $N = 6$ | 768 | 3072 | 12 | 200M 209M | En-De En-Fr | **28.67** **43.40** |

### B.3 ROBUSTNESS OF THE NUMBER OF COLUMNS

In the ablation analysis of the paper, we show that when fix the total FLOPs and add more columns of RevCol, the performance first increases and then get saturated. When the number of columns is extreme large, such as 20, the performance drop because of the representation ability of single column is limited. When the number of columns is usual, such as 4˜12, the performances are similar, which verifies the setting robustness of the number of columns.

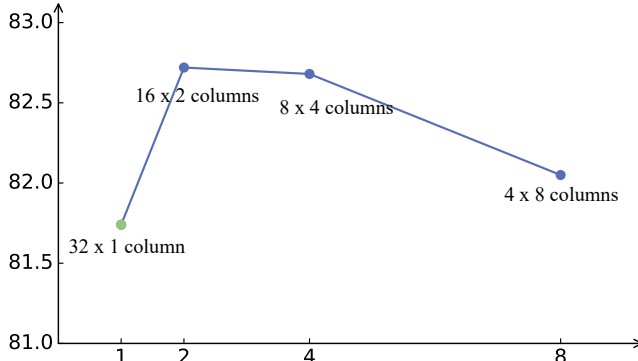

Figure 4: ImageNet top-1 accuracy of different variants of RevCol-ViT-B. Each variant has the same total number of residual blocks and channel dimension.

To further analyze the robustness of the number of columns, in this section, we build some RevCol-ViT-B variants (see Appendix B for more details). Each variant has the same number of residual blocks with the same channel dimension, but different number of columns. In other worlds, these variants have the same channel dimension and different depth of each columns and different number of columns. We use 32 residual blocks totally and maintain the FLOPs about 18G. Fig. 4 show the performance on ImageNet-1K of different variants. The number of columns are 1, 2, 4, and 8, accordingly the depth of each column are 32, 16, 8, and 4. The performance of single column variant is lower (similar to DeiT-B (Touvron et al., 2020)) because of the single column ViT can not maintain the information as multi reversible columns. The performance is decreasing when the number of columns became larger, because of the depth of each columns is not enough. This phenomenon indicates us that given a target FLOPs, the setting of the number of columns is robust unless the depth of each columns or channel dimension is too small.

## C SYSTEM-LEVEL COMPARISON WITH SOTA FOUNDATION MODELS

Foundation models (Kolesnikov et al., 2020; Radford et al., 2021; Yuan et al., 2021b) are general-purpose backbones pre-trained on massive and divergent data source. They can adapt to various down-stream tasks with limited domain-specific data. We show comparison among various public *state-of-the-art (SOTA)* foundation models including Vision Transformers and Vision-Language models, namely, *SwinV2* (Liu et al., 2022a), *BEiT3* (Wang et al., 2022), and *Florence* (Yuan et al., 2021b). As shown in Tab. 6, though our RevCol-H is *purely convolutional and pre-trained on single modality dataset*, the results on different tasks demonstrate remarkable generalization ability of RevCol with large scale parameters.

Table 6: **System-level comparison of state-of-the-art visual foundation models with large-scale pretraining.** We include ○ Vision Transformers, ● CNNs, and ● hybrid architectures pretrained either *unsupervised* or *supervised* on *image-only* and *vision-language* datasets. COCO scores marked with † means intermediate fine-tuned on extra data like Object365 (Shao et al., 2019).

| Model | Params | Dataset | | ImageNet | COCO test-dev | | | ADE20K | | |
| | | Images | Annotation | 1k | Detector | $AP_{box}$ | $AP_{mask}$ | Segmenter | mIoU | +ms |
|---|---|---|---|---|---|---|---|---|---|---|
| ○ SwinV2-G | 3.0 G | 70 M | labeled | 90.2 | HTC++ | 63.1† | 54.4† | UperNet | 59.3 | 59.9 |
| ○ BEiT3 | 1.0 G | 35 M | labeled & image-text | 89.6 | ViTDet | 63.7† | 54.8† | Mask2Former | 62.0 | 62.8 |
| ● Florence | 0.9 G | 900 M | image-text | 90.1 | DyHead | 62.4 | - | - | - | - |
| ● RevCol-H | 2.1 G | 168 M | semi-labeled | 90.0 | DINO | 63.6† | - | Mask2Former | 60.4 | 61.0 |

# D MORE ANALYSIS EXPERIMENTS

## D.1 PERFORMANCE GAIN OF REVERSIBLE COLUMNS ARCHITECTURE

In this section, we evaluate the performance gain of using reversible columns. In the first experiment, we fix a single column's structure and FLOPs then simply add more columns to scale large and test the performance. At the same time, we plot the vanilla single-column models with similar model sizes. As depicted in Fig. 5, compared to single-column models, using multi-column reversible architecture always gets better performance under same FLOPs constraint. Besides, within a certain range, scaling up RevCol in terms of increasing column numbers can have similar gains compared to scaling up with both block numbers(depth) and channel numbers(width) in single-column models. In the second experiment, we limit the model size to about 4.5G FLOPs and test model variants with different column numbers. In other words, we gradually add more columns and scale down the single column size at the same time. Results are shown in Tab. 7, we notice that adopt column number at the range of 4 to 12 can maintain the model's performance, then further more column models suffer from performance degradation. We believe the reason is the width and depth in a single column are too low to keep representation ability.

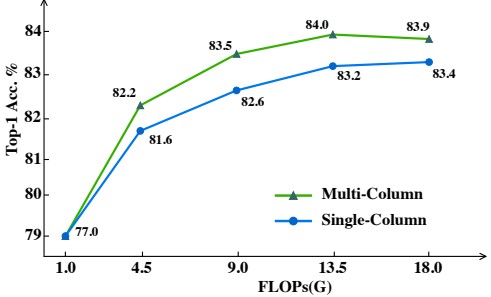

Figure 5: ImageNet-1K performance of maintaining a constant FLOPs of a single column and adding more columns.

Table 7: ImageNet 1K performances of various number of columns in RevCols under the similar computational budget.

| # column | Params | FLOPs | FLOPs per col. | Top-1 Acc. |
|---|---|---|---|---|
| 1 | 28M | 4.4G | 4.40G | 81.9 |
| 4 | 30M | 4.5G | 1.12G | 82.2 |
| 8 | 34M | 4.7G | 0.59G | 82.3 |
| 12 | 33M | 4.4G | 0.35G | 82.2 |
| 20 | 35M | 4.2G | 0.21G | 81.0 |

## D.2 REVERSIBLE NETWORKS VS. NON-REVERSIBLE NETWORKS

In this section, we ablate different design patterns of reversible connections. First, we build a non-reversible multi-column network using the fusion module of HRNet. Second, we build another single column reversible ConvNeXt using the design of RevNet as shown in Fig. 2(a). We compare the two designs with our RevCols. The evaluation result is shown in Tab. 8. The non-reversible multi-column network suffers from information loss during propagation, which could result in lower accuracy. The reversible single-column network maintains information during propagation, but lack the superiority of multi-level fusion. This experiment further indicates the effectiveness of combining the reversible design with multi-column networks.

Table 8: Performance comparison on ImageNet-1K of different design patterns. Row-1 represents HRNet style network w/o *reversible connections*. Row-2 represents RevNet style network w/o *multi-column fusions*. Row-3 are our proposed RevCols.

| rev. conn. | multi-col. | Params | FLOPs | Acc. |
|:---:|:---:|:---:|:---:|:---:|
|  | ✓ | 35M | 4.9G | 78.8 |
| ✓ |  | 34M | 4.5G | 81.6 |
| ✓ | ✓ | 30M | 4.5G | 82.2 |

Table 9: Performance comparison between models with and without *intermediate supervision*. Results are reported on ImageNet-1K and COCO dataset. We use 1× training schedule on COCO detection task.

| Model | inter. sup. | Top-1 Acc. | $AP_{box}$ | $AP_{mask}$ |
|:---|:---:|:---|:---|:---|
| RevCol-T | ✗ | 81.4 | 48.3 | 41.8 |
| RevCol-T | ✓ | 82.2 (+0.8) | 48.8 (+0.6) | 42.2 (+0.4) |
| RevCol-S | ✗ | 83.0 | 50.7 | 43.8 |
| RevCol-S | ✓ | 83.5 (+0.5) | 51.1 (+0.4) | 43.8 (+0.0) |
| RevCol-B | ✗ | 83.2 | 51.2 | 44.2 |
| RevCol-B | ✓ | 84.1 (+0.9) | 51.6 (+0.4) | 44.2 (+0.0) |

## D.3 PERFORMANCE GAIN OF USING INTERMEDIATE SUPERVISION

In this section, we evaluate the performance of RevCol-T/S/B with and without intermediate supervision on ImageNet-1K. We also evaluate the object detection task performance using 1× training schedule on MS-COCO dataset. Other settings remain the same. From the validation results in Tab. 9, models trained with intermediate supervision achieves 0.5% to 0.9% better top-1 accuracy. Besides, intermediate supervision also benefits down-stream tasks, which further demonstrates its effectiveness.

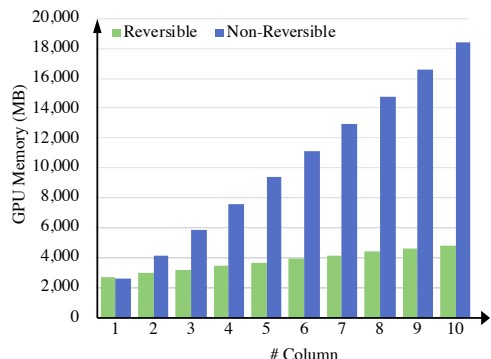

Figure 6: GPU Memory Consumption *vs*. Model size

Table 10: Performance of models with larger kernel convolution.

| Kernel Size | FLOPs | Top-1 Acc | $AP_{box}$ 1× | $AP_{mask}$ 1× |
|:---|:---:|:---:|:---:|:---:|
| 3 | 4.5G | 82.2 | 48.8 | 42.2 |
| 5 | 4.5G | 82.5 | 49.5 | 42.6 |
| 7 | 4.6G | 82.5 | 49.3 | 42.4 |
| 11 | 4.6G | 82.5 | 49.9 | 42.7 |

## D.4 GPU MEMORY CONSUMPTION VS MODEL SIZE

Fig. 6 plots the GPU memory consumption with the scaling of model size. We fix the computation complexity of a single column to 1G FLOPs and increase column number. Meanwhile, we measure the memory consumption in training process which includes the forward and backward propagation. Our experiments are conducted on Nvidia Tesla V100 GPU under batch-size 64, FP16 precision and PyTorch implementation. With the increment of column number, we can see RevCol keeps an $\mathcal{O}(1)$ GPU memory consumption, while non-reversible architecture's memory consumption increase linearly with column number. Note that our RevCol does not keep strictly the same GPU memory consumption as column number increase, as reversible networks need to back-up the operation weights in need for calculating gradients and the re-construction of feature maps in backward propagation.

## D.5 ABLATION OF KERNEL SIZE IN CONVOLUTIONS

In original ConvNeXt, large kernel convolution achieves in better performance. We conduct experiments in RevCol-T. As shown in Tab. 10, for 4 column models, using $5 \times 5$ convolution increase the ImageNet-1k Top-1 accuracy by 0.3% and the COCO $AP_{box}$ by 0.7 for RevCol-T model. Further increasing kernel size obtains more accuracy in down-stream tasks, but not too much. We consider the RevCol design already enlarges the effective receptive field and this limit the accuracy gain of using large kernel convolution. On the other hand, $3 \times 3$ convolution enjoys the merits of efficiency and stability in (pre)training. Therefore, we adopt kernel 3 in all RevCol models.

## E    SEMI-LABELED PRIVATELY COLLECTED DATASET FOR LARGE MODELS

### E.1    DATA COLLECTION AND PSEUDO LABEL SYSTEM

The dataset consists of around 168 million(M) images, 50M of which labeled and the remaining 118M unlabeled. The majority of labeled images come from public datasets, *e.g*. ImageNet, Places365 (Zhou et al., 2017a), and Bamboo (Zhang et al., 2022b). The others are web-crawled images annotated by in-door employees. Unlabeled images come from weakly-annotated image-text datasets like YFCC-100M (Thomee et al., 2016). We do not use text annotations.

In order to utilize images of different label domains and the massive unlabeled images, we employ a multi-target label system similar to Ding et al. (2022a) and Ghiasi et al. (2021). We adopt a semi-supervised learning strategy with ViTs, thus generating *pseudo* labels with continuously increased quality. We only store soft predictions with confidence higher than 1% to save storage. The final version of *pseudo* label we use are generated by a multi-head ViT-Huge teacher, which has an 89.0% ImageNet-1k accuracy.

### E.2    IMAGE DEDUPLICATION

Since the dataset contains large amount of unverified web-crawled images, there are probably validation or test images sneaking into our training dataset. Works like Mahajan et al. (2018) and Yalniz et al. (2019) all regard image deduplication an important procedure for fair experiments.

We first iterate over the entire dataset to filter out suspicious duplicates together with the corresponding test images based on their *pseudo* label distance. This brings more than 10,000 images with high suspicion. We look at these image pairs and finally find about 1,200 exact-duplicates and near-duplicates. Fig. 7 shows some examples of the near-duplicates, which are difficult to detect. Never the less, training a model without removing these duplicates gives less than 0.1% accuracy gain on ImageNet-1k in our experiments. We attribute this to the absence of true labels from these duplicates.

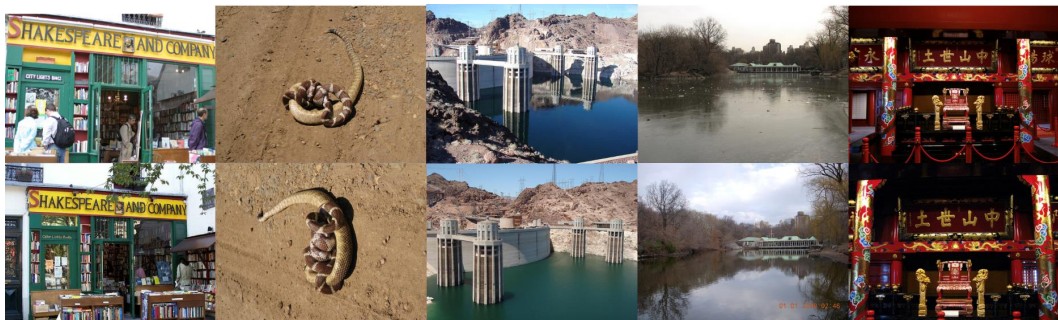

Figure 7: Top: Near duplicates found in unlabeled images. Bottom: ImageNet-1k validation images.

## F    MORE TRAINING DETAILS

This section gives more training details on ImageNet classification, COCO detection, and ADE20K segmentation.

### F.1    INTERMEDIATE SUPERVISION SETTINGS

We add intermediate supervision in ImageNet-1k training, ImageNet-22k and extra data pre-training. We used a 3-block decoder with gradually up-sampled feature maps in ImageNet-1k training. The block setting remains the same as Sec. 2.2 We use a single layer decoder in ImageNet-22k and extra data pre-training. For all the variants of RevCol, we set the number of compound loss $n$ to 3 empirically (eg. for a 8 column RevCol, the intermediate supervision is added to column 2, 4, and 6, and the original classification CE loss is also added to column 8). $\alpha_i$ is set to 3, 2, 1, 0 and $\beta_i$ is set to 0.18, 0.35, 0.53, 1.

## F.2 Hyperparameters Used for Training and Pre-training

This section introduces the training details for main experiments, the supervised training on ImageNet and extra data. We show this setting in Tab. 11. All experiments in ablation studies are superivised trained on ImageNet-1K except additional descriptions and also follow settings described in this section.

Table 11: Hyperparameters for training and pre-training RevCol.

| Hyperparameters | ImageNet-1K | ImageNet-22K | 168M Extra Data |
| --- | --- | --- | --- |
| | T/S/B | B/L/XL | XL/H |
| Input resolution | $224^2$ | | $224^2$ |
| Training epochs | 300 | 90 | 10 |
| Warmup epochs | 20 | 5 | 0.15 |
| Batch size | 4096 | | 5120 |
| Peak learning rate | 4e-3 | 5e-4 | 6.25e-4 |
| Learning rate schedule | cosine | | cosine |
| Layer-wise learning rate decay | ✗ | | ✗ |
| AdamW momentum | (0.9, 0.999) | | (0.9, 0.999) |
| Weight decay | 0.05 | 0.1 | 0.05 |
| Gradient clipping | ✗ | | 1.0 (element-wise) |
| Drop path | 0.1/0.3/0.4 | 0.3 | 0.2 |
| EMA | 0.9999 | ✗ | ✗ |
| Label smoothing $\varepsilon$ | 0.1 | | 0.1 |
| Data augment | RandAug (9, 0.5) | | RandAug (9, 0.5) |
| Mixup | 0.8 | | ✗ |
| CutMix | 1.0 | | ✗ |
| Random erase | 0.25 | | ✗ |

## F.3 Hyperparameters Used for Fine-tuning

This section gives the hyperparameters used for fine-tuning on ImageNet-1K and downstrea COCO object detection and instance segmentation, ADE20K semantic segmentation tasks, as shown in Tab. 12, Tab. 13 and Tab. 14.

Table 12: Hyperparameters for fine-tuning RevCol on ImageNet-1K classification.

| Hyperparameters | ImageNet-1K |
| --- | --- |
| | B/L/XL/H |
| Input resolution | $384^2/384^2/384^2/640^2$ |
| Fine-tuning epochs | 30 |
| Warmup epochs | 0 |
| Batch size | 512 |
| Peak learning rate | 5e-5 |
| Layer-wise learning rate decay | 0.9/0.8/0.8/0.8 |
| AdamW momentum | (0.9, 0.999) |
| Weight decay | 1e-8 |
| Learning rate schedule | cosine |
| Head init scale | 0.001 |
| Drop path | 0.2/0.3/0.4/0.5 |
| EMA | ✗/✗/✗/0.9999 |
| Gradient clipping | 10.0 (norm) |
| Label smoothing $\varepsilon$ | 0.1 |
| Data augment | RandAug (9, 0.5) |
| Mixup | ✗ |
| CutMix | ✗ |
| Random erase | 0.25 |

Table 13: Hyperparameters for fine-tuning RevCol on object detection with Cascade Mask R-CNN detector.

| Hyperparameters | IN-1K Pre-trained | IN-22K Pre-trained |
| --- | --- | --- |
| | RevCol-T/S/B | RevCol-B/L |
| Fine-tuning epochs | 36 | |
| Batch size | 16 | |
| Peak learning rate | 2e-4 | 1e-4 |
| Warmup steps | 1500 | |
| Layer-wise learning rate decay | 0.85/0.8/0.8 | 0.9/0.8 |
| AdamW momentum | (0.9, 0.999) | |
| Weight decay | 0.05 | |
| Drop path | 0.3/0.4/0.4 | 0.5/0.6 |

Table 14: Hyperparameters for fine-tuning RevCol on ADE20K semantic segmentation with UperNet segmentation framework.

| Hyperparameters | IN-1K Pre-trained | IN-22K Pre-trained |
| --- | --- | --- |
| | RevCol-T/S/B | RevCol-B/L |
| Input resolution | $512^2$ | $640^2$ |
| Fine-tuning steps | 80k | |
| Batch size | 16 | |
| Peak learning rate | 4e-5 | |
| Warmup steps | 1500 | |
| Layer-wise learning rate decay | 1.0 | 0.9 |
| AdamW momentum | (0.9, 0.999) | |
| Weight decay | 0.01 | |
| Drop path | 0.3 | |

### F.3.1 CONVOLUTION KERNEL PADDING TRICK IN DOWN-STREAM TASKS

According the results shown in Section D.5, larger kernel convolution perform better especially in down-stream tasks. To save the pre-training cost meanwhile achieve better performance, we pad the small $3 \times 3$ convolution kernel in pre-trained model weights to larger size then fine-tune in detection and segmentation tasks. Inspired by *Net2net* (Chen et al., 2015) method, we pad the pre-trained $3 \times 3$ kernel in convolution layer with Gaussian initialized values. To protect the pre-trained kernel from being disturbed by the new padded values, we initialize the padded values with 0 mean and extremely small standard deviations (1e-7). We use this trick only with our largest model RevCol-H. We pad the $3 \times 3$ kernel in pre-trained model to $7 \times 7$ kernel size in COCO detection task and $13 \times 13$ in ADE20k sementatic segmentation task, then fine-tune on corresponding dataset to get the final result. In general, the kernel padding trick leads to 0.5~0.8 $AP_{box}$ improvement and 0.7~1.0 mIoU improvement for RevCol-H model.

## G   VISUALIZATIONS OF FEATURE DISENTANGLEMENT

In this section, we show our RevCol can disentangle features with stacked columns, which is different from the conventional sequential networks. We use RevCol-S pre-trained on ImageNet-1K for analysis. First, we visualize the class activation maps (CAMs) for outputs of each last layer of a level. We adopt LayerCAM (Jiang et al., 2021) technology to generate the CAMs with the predicted classes. Fig. 8 show the heatmaps of activation. With the levels and columns going deeper, the features focus on the regions with more semantics. The outputs of RevCol-S are the different levels of last column. These features with high level semantics focus on different parts of the image and the whole part of the object, achieving disentanglement of features for task-relevant and task-irrelevant.

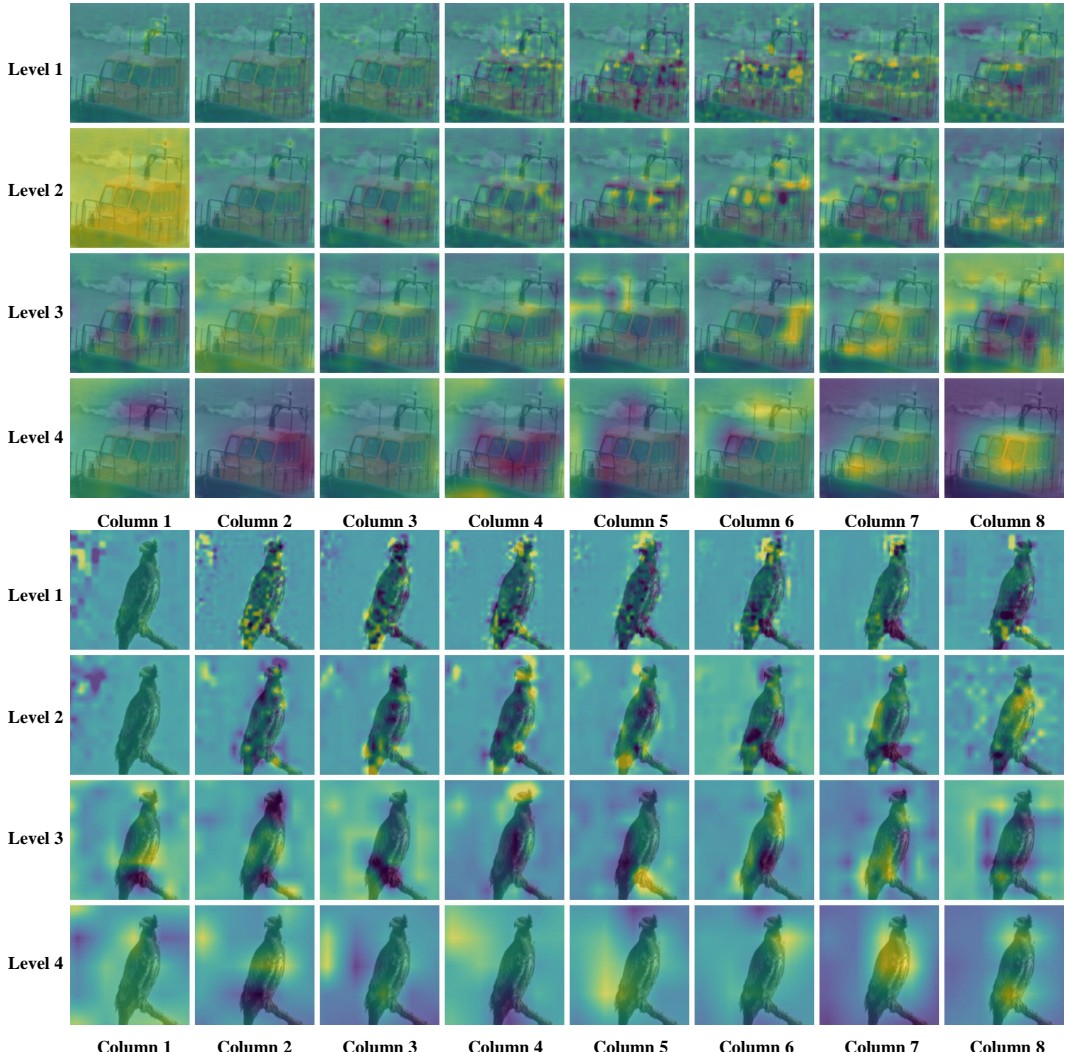

Figure 8: Visualizations of class activation maps using LayerCAM (Jiang et al., 2021) for different levels and columns.

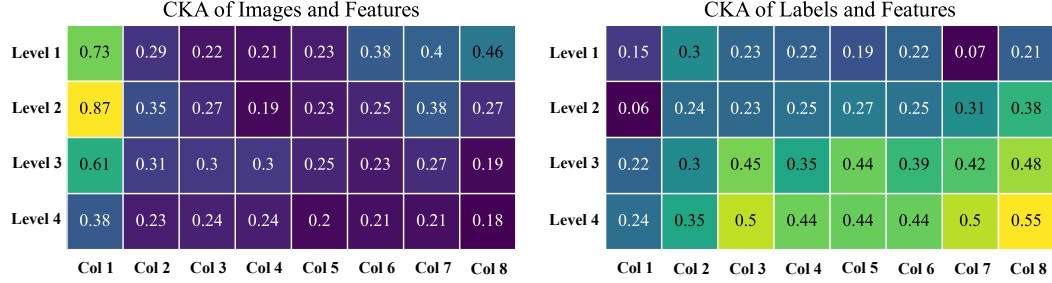

Figure 9: CKA similarities (Kornblith et al., 2019) of features and images/labels for different levels and columns.

To quantify the disentanglement, we use Centered Kernel Alignment (CKA) similarity metric (Kornblith et al., 2019) to measure the similarity between representations in RevCol-S. We calculate the CKA similarities between intermediate features in different levels and columns and images or labels of each category in the ImageNet val set. Then we plot the similarities of the category with

the highest label similarity in Fig. 9. As shown in the figure, the similarities between images and intermediate features are not clearly distinguished at different levels in Column 2-5, while the features with higher levels have lower similarity to the images in Column 6-8. The similarities between labels and intermediate features are also more distinct in higher columns.

