# OpenReview forum: "Reversible Column Networks"
_ICLR.cc/2023/Conference — ICLR 2023 poster_

### Official Review · Reviewer_cXh3 · 2022-10-20

**Confidence:** 3
**Correctness:** 3
**Technical Novelty And Significance:** 3
**Empirical Novelty And Significance:** 3
**Recommendation:** 6

**Clarity, Quality, Novelty And Reproducibility:**

The paper is generally written well, but it would be better if figures/tables could be arranged to where they are referred to for the first time. In the network design part, the paper sometimes simply describes the design choices without explanations of motivations and justifications or puts them far away. The paper describes its key designs clearly for reproduction. Personally, I feel that it is interesting to combine GLOM architecture with reversible transformations, and this is the key contribution.

**Strength And Weaknesses:**

Strong points.
1. The paper introduces a new CNN-based design with reversible transformations and GLOM architecture, which can inspire future research.
2. The experiments show the competitive performance of the proposed method on several vision tasks.
3. The paper provides detailed information for reproduction.

Weak points.
1. The paper sometimes draws conclusions without enough contextual information/justifications or leaves analysis far away from the conclusions, which makes it difficult for readers to understand, especially in the macro and micro design sections.

2. The paper does not differentiate itself clearly from existing methods, especially from (Hinton, 2021; Wang et al. 2021) and (Chang et al. 2018; Jacobsen et al. 2018; Mangalam et al. 2022).

Minor issues.
Line 9 Section 1, "Researches" -> "Researchers"
The proposed method is sometimes called "RevCols" but sometimes called "RevCol".

Questions.
1. The difference between the proposed method and GLOM is not clear. The "Relation to previous works" only mentions one difference but does not explain the reason and advantages/disadvantages.
2. Sec. 2.2.1 simplifies the multi-column reversible of Sec. 2.1. Would this still satisfy the reversible criterion?
3. Will code be released?

**Summary Of The Paper:**

This paper proposes a new neural network architecture design based on GLOM (Hinton, 2021) called the Reversible Column Networks (RevCols). RevCols has multiple sub-networks (called columns) of the same structure, between which multi-level reversible units are added. The paper extends RevNets (Gomez et al., 2017) in a recursive way, then splits features into groups and reorganizes groups as columns. The paper leverages two auxiliary heads (Wang et al. 2021) to help disentangle the features. Experiments show that RevCols achieves competitive performances on several vision tasks including image classification, object detection, and semantic segmentation tasks.

**Summary Of The Review:**

I am on the positive side of this paper. I think it is promising to combine GLOM architecture with reversible transformations. However, I am confused but interested to know the motivation behind it and the detailed analysis between these two works. I am happy to increase my rating if the authors can address this ambiguity.

---

> ### Author Response · Authors · 2022-11-18
> **Response to Reviewer cXh3 Part 1**
>
> Dear Reviewer cXh3,
>
> Thank you for your valuable feedback. We will address the concerns and questions individually and answer them below.
>
>
> 1. **"The paper sometimes draws conclusions without enough contextual information/justifications or leaves analysis far away from the conclusions, which makes it difficult for readers to understand, especially in the macro and micro design sections."**
>
> - Thanks for pointing out the unclear issue. We have re-written the paper, especially the sections on macro and micro design for more clarity.
>
> 2. **"The paper does not differentiate itself clearly from existing methods, especially from (Hinton, 2021; Wang et al. 2021) and (Chang et al. 2018; Jacobsen et al. 2018; Mangalam et al. 2022)."**
>
> 3. **"The difference between the proposed method and GLOM is not clear. The "Relation to previous works" only mentions one difference but does not explain the reason and advantages/disadvantages."**
>
> 4. **"I think it is promising to combine GLOM architecture with reversible transformations. However, I am confused but interested to know the motivation behind it and the detailed analysis between these two works. "**
>
> Answer 2-4:
>
> **The motivation of combine GLOM with reversible transformations:**
>
>   As early as 1990s, Hinton proposed features in neural network should learn the part-whole hierarchies. However, modern neural networks follows the IB rules, which gradually compress the information unrelated to target. In GLOM, Hinton revisit this goal and depict a big picture about parsing the visual scenes into part-whole hierarchies. Features of a neural network should embed the visual input from object level to scene level gradually. In other words, representation lies in a disentangled manner, from the part to the whole. GLOM use a time sequence based method to gradually settle down the representation in feature dimensions, which is also referred as formulating islands.
>
>   The forward-looking insight of network make it hard to implement with current operations and optimization techniques. For example, In GLOM's time sequence based method, it cannot keep information from being lost. To avoid feature collapse, GLOM suggests contrastive auxiliary loss. Contrastive training methods need extra pairs of positive and negative samples, which is complicated and unstable. In RevCol, we use reversible transformations between columns to provides by nature lossless information propagation.
>   To be specific, the network store multi-level embeddings, i.e. from low-level to highly semantic representations in feature dimensions, and the whole output features maintain as much information as input. In this way, the high-level semantic representations are tuned during ImageNet classification pre-training and other low-level information is also maintained in other feature dimensions in case of the demand of down-stream tasks.
>
>
>
>   Let us revisit the process of building RevCol from RevNet, and **we will state the motivation behind each part's design and the difference between RevCol and other architectures**.
>
>   Figure.1 (https://i.ibb.co/rb5h6T3/figure1.png) is a sketch of Reversible Unit in RevNet. And we also include the forward and inverse computation in paper **Eq. 1.**
>
> $$
> Forward: x\_{t} = \\boldsymbol{F}_t(x_\{t-1}) + \\gamma x\_{t-2} $$
>
> $$  Inverse: x\_{t-2} = \\gamma^\{-1}[x_{t} - \\boldsymbol{F}\_t(x_{t-1})]
> $$
>
> $F_t$ denotes an arbitrary non-linear operation analogous to those residual functions in standard ResNets; $\gamma$ is a simple reversible operation (e.g. channel-wise scaling), whose inverse is denoted by $\gamma^{-1}$.
>
>   The **advantages** of RevNet:
>
>   1. Feature $x_{t-2}$ can be reconstructed through the inverse computation in Eq.1 and this computation only relies on features $x_{t-1}$ and $x_{t}$. No more features are needed, so that the peak memory consumption is greatly reduced.
>
>   2. Information keeps lossless propagation in forward process, as the initial input {x1, x2} can be reconstructed after later calculations.
>
>   The **defects** of RevNet:
>
>   1. Eq.1 involves too strong constraint on the feature dimensions, i.e. $x_{t}, x_{t+2}, x_{t+4}, ...$ have to be **equal sized**, which is not flexible in architecture design.
>
>   2. The last two feature maps in RevNet have to contain both low-level and semantic information due to the reversible nature, which may be difficult to optimize as in conflict with IB principle. (e.g. In general, feature close to loss contains more semantic information in classification task)

---

> > ### Author Response · Authors · 2022-11-18
> > **Response to Reviewer cXh3 Part 2**
> >
> > **Our insight on Eq.1:**
> >
> > 1. Eq.1 does not require feature maps of each item to be equal sized. In detail, to calculate feature $x_{t}$,  $x_{t-2}$ need to be in the same shape of $x_{t}$. While such constraint does not exist on $x_{t-1}$. The non-linear operations in $\\boldsymbol{F}\_t ()$ could map feature $x_{t-1}$ to be of the shape with $x_{t}$. After that, we can simply do the summation.
> >
> > 2. Eq. 1 does not limit the number of inputs to be strictly 2. Even if we add more inputs to item $\boldsymbol{F}_t ()$, the reversible character are still satisfied. What we need to do in backward process is to reconstruct these inputs with Eq. 1 Inverse, and then recalculate item $\boldsymbol{F}_t ()$.
> >
> >   **How we tackle the defects of RevNet:**
> >
> >   Based on the insight above, we add more inputs of different shapes to item $\boldsymbol{F}_t()$. Therefore, we can use tensors of different shape to represent features of different semantic levels or different resolutions. We generalize Eq. 1 into the following **Eq.2** in paper:
> >
> >   $$
> >   Forward: x_{t} = \\boldsymbol{F}\_t(x\_{t-1}, x\_{t-2}, ..., x\_{t-m+1}) +  \\gamma x\_{t-m}
> >   $$
> >   $$
> >   Inverse: x_{t-m} = \\gamma^{-1}[x\_{t} - \\boldsymbol{F}\_t(x\_{t-1}, x\_{t-2}, ..., x\_{t-m+1}) ]
> >   $$
> >
> >   m is the order of the recursion (m $\ge$ 2). Then we partition every $m$ feature maps into a group: $(x_1, x_2, \dots, x_m), (x_{m+1}, x_{m+2}, \dots, x_{2m}), \dots$. Given the features within any of the group, we can easily compute the features in other groups recursively according to Eq. 2. By reorganizing these groups into a *multi-column* fashion, the network can propagate reversibly, as Figure 2 (https://i.ibb.co/6vK6VZ6/figure2.png) shows.
> >
> >   By now, hope you can understand the modifications we made and **the difference to previous RevNet series of works** (Chang et al. 2018; Jacobsen et al. 2018; Mangalam et al. 2022).
> >
> >   **The difference to GLOM (Hinton, 2021):**
> >
> >   GLOM interprets the scene parsing of neural network as formulating islands. Towards this goal, it require each level to connect to its neighbor levels in **previous** column. In RevCol, we need to obey the rule of reversible, so the connections lies in different manner. We depict this difference in Figure 3(https://i.ibb.co/WnQ0kny/figure3.png). Besides, GLOM has lots of special design for formulating islands. For example, the columns are weight-sharing; Each Level's building block is a spatially local auto-encoder, only 1x1 convolutions are used; Interactions between locations are done by attention-weighted average of the embeddings. In RevCol, the general design of reversible column can integrate many different building block, including CNN and vision transformer. Currently we do not design a special building block for RevCol architecture.
> >
> >
> >
> > 5. **"Sec. 2.2.1 simplifies the multi-column reversible of Sec. 2.1. Would this still satisfy the reversible criterion?"**
> >
> > - Yes, it's still reversible. In **insight 2** above, we mention reversible do not have limitations on the number of inputs to $\\boldsymbol{F}_t()$. Our simplification in Sec. 2.2.1 reduce inputs of $\\boldsymbol{F}_t()$ from m-1 to 2.
> >
> > 6. **"Will code be released?"**
> >
> > - Yes. Codes and model weights will be released very soon.

---

### Official Review · Reviewer_BtZK · 2022-10-23

**Confidence:** 4
**Clarity, Quality, Novelty And Reproducibility:** The writing is good and clear. There …
**Correctness:** 3
**Technical Novelty And Significance:** 2
**Empirical Novelty And Significance:** 3
**Recommendation:** 6

**Strength And Weaknesses:**

The experiments are conducted among many areas, including image classification, object detection, semantic image segmentation and machine translation. Through these experiments, I feel that there are some kinds of competitiveness in this method, but not many.
For example, in Table 1, it shows this method behaves consistently worse than EfficientNet, and SwinV2 is missing. In Table 9, it shows this method behaves consistently worse than SwinV2.

**Summary Of The Paper:**

This paper is mainly about convolutional neural networks for image classification. The authors propose a new paradigm to fuse features from different depths. The are some parallel branches of convolutional layers. Features from each branch are fused through a complex topology as stated in Figure 2(c). Some deep supervised loss are adopted for some branches.

**Summary Of The Review:**

Abound experiment are designed, on many topics, and The results of ablation study of feature fuse and deep supervision support the conclusion.

---

> ### Author Response · Authors · 2022-11-18
> **Response to Reviewer BtZK**
>
> Dear Reviewer BtZK,
>
> Thank you for your valuable feedback. We will address the concern of experiments and answer it below.
>
> **"Through these experiments, I feel that there are some kinds of competitiveness in this method, but not many. For example, in Table 1, it shows this method behaves consistently worse than EfficientNet, and SwinV2 is missing. In Table 9, it shows this method behaves consistently worse than SwinV2."**
>
> - We understand your concerns about the absolute performance. We answer your doubts in the following points:
>
>   1. RevCol is not specifically designed to pursue SOTA performance. On the contrary, it is a general architecture that maintains lossless information during propagation. Opposed to conventional networks, this property enables RevCol to learn disentangled features through reversible column propagation (As shown in Fig. 1 in the paper). With given building blocks (such as depth-wise convolution based ConvNeXt block and self-attention based Transformer block), RevCol can build upon it to learn disentangled feature.
>
>   2. Given a certain FLOPS or parameters budget, it can be expected that disentangled feature learning may be not as good as the conventional counterparts benchmarked with **the only one dataset**, because disentangled feature learning has to cost some computations on some unused features. We argue that disentangled learning is potential to have better transfer learning abilities. For example, the potential advantages in low-level vision tasks and domain transfer tasks.
>
>   3. We do not list the results of SwinV2 because all variants of SwinV2 have larger FLOPs than and similar works, which makes it hard to make a fair comparison. The EfficientNet design, including kernel size, expansion, width, and depth are task-oriented network architecture searched results. It needs specifically changes or additional head designing when transfer to downstream tasks. In RevCol, our goal is to design a general architecture that learns disentangled feature. The macro design of RevCol also generalizes well to Transformer building blocks and NLP tasks. Besides, on ImageNet classification, RevCol behaves NOT consistently worse than EfficientNet series, such as RevCol-B with lower FLOPs performs better than EfficientNet-B6 (84.1 vs. 84.0), as shown in Tab. 1 in the paper.
>
>      We adopt convolutional based building block by default in the main paper and show its promising results. In the appendix, we also show RevCol architecture can generalize to self-attention based Transformer building block with higher performance than vanilla baseline. We experiment by simply replacing the basic blocks with window-attention block adopted in SwinV2. The base size of RevCol-SwinV2 model can learn disentangled features and also archieves 84.1% top-1 accuracy on ImageNet classification using the same training setting as SwinV2.
>
>   4. We adopt 3x3 convolution kernel in the paper for efficiency and simplicity. However, ConvNeXt adopts 7x7 convolutional kernel size by default and shows the advantages (especially on downstream tasks that require large receptive fields) in their paper. We can also use a larger kernel size (7x7) to achieve higher performance which is a fairer comparison to ConvNeXt. In our experiments, the smallest model RevCol-T with 7x7 kernel achieve **82.3 Top-1 accuracy on ImageNet-1k (+0.1)**, **51.0 box AP on COCO detection (+0.4)**. With ImageNet-22k pre-training, RevCol-B with 7x7 achieve **86.8% Top-1 Accuracy on ImageNet-1k (+0.1)** , **54.3 box AP on COCO detection (+0.8)**.
>
>   5. RevCol explores a new scale up dimension, column number, in addition to model depth and channel width. Increasing column numbers has similar income as increasing both width and depth in certain range. With the new scaling dimension, our largest model RevCol-H achieves the best COCO detection (63.1 box AP) and ADE20K segmentation (61.0 mIoU) result among pure (static) CNN models. (Results with DINO framework on COCO detection and Mask2Former framework on ADE20K segmentation were updated in Tab.1 and Tab.2 in the paper.) This verifies the scalability of the reversible column network architecture which can extend the model upper bound.

---

### Official Review · Reviewer_AtgT · 2022-10-24

**Confidence:** 3
**Correctness:** 3
**Technical Novelty And Significance:** 3
**Empirical Novelty And Significance:** 3
**Recommendation:** 6

**Clarity, Quality, Novelty And Reproducibility:**

* **Clarity**: most contents are clear and easy to follow.
* **Quality**: The proposed method is evaluated and studied on substantial benchmarks and settings.
* **Novelty**: The design principles have some in common with previous methods, but some details are new.
* **Reproducibility**: Most implementation details are provided but the code is not available.

**Strength And Weaknesses:**

**Strength**
* It is important to explore strong and disentangled representations by designing new neural architectures. The proposed reversible column manner is reasonable to maintain both high- and low-level information.
* The proposed method shows promising experimental performance.

**Weaknesses**
* It is expected to explain the intrinsic difference between the proposed method and previous ones, including but not limited to HRNet and NAS (neural architecture searched) networks. Especially, networks [1,2] searched in the cell-based/-like search space have many in common with the proposed one. A more comprehensive analysis is suggested, not only the provided demonstration from the motivation perspective.
* The shown experimental performance is competitive but the promotion over previous ones is not so evident. I admit the number is almost saturated, but any other advantage this method could bring needs to be demonstrated.
* Real throughput/latency needs to be measured to more accurately validate the model budget, not just FLOPs or #Params. The introduced connections seem to introduce larger latency on real hardware which is not so related to FLOPs numbers.


[1] Liu C, Chen L C, Schroff F, et al. Auto-deeplab: Hierarchical neural architecture search for semantic image segmentation[C]//Proceedings of the IEEE/CVF conference on computer vision and pattern recognition. 2019: 82-92.
[2] Xie S, Kirillov A, Girshick R, et al. Exploring randomly wired neural networks for image recognition[C]//Proceedings of the IEEE/CVF International Conference on Computer Vision. 2019: 1284-1293.

**Summary Of The Paper:**

This paper proposes a new neural network paradigm, which aims at gradually disentangling features in the forward propagation. The whole network is constructed with multiple copies of sub-networks. The total information can be maintained rather than compressed during propagation. The proposed method is also evaluated on several typical computer vision tasks and shows competitive performance.

**Summary Of The Review:**

The overall idea is good while the main concerns lie in the demonstration of advantages or differences with previous methods and speed analysis.

---

> ### Author Response · Authors · 2022-11-18
> **Response to Reviewer AtgT Part 1**
>
> Dear Reviewer AtgT,
>
> Thank you for your valuable feedback. We will address concerns individually and answer them below.
>
> 1. "**It is expected to explain the intrinsic difference between the proposed method and previous ones, including but not limited to HRNet and NAS (neural architecture searched) networks. Especially, networks [1,2] searched in the cell-based/-like search space have many in common with the proposed one. A more comprehensive analysis is suggested, not only the provided demonstration from the motivation perspective."**
>
> - Let us firstly give a comprehensive understanding about the proposed RevCol and NAS based networks. Modern NAS algorithms heavily depends on the search space which relies on human. The basic cell/unit/connection  in search space come form human-designed operators, such as convolution, BN, residual connection etc. Even researchers[1] try to search from scratch (basic arithmetic), the result in network still not suitable for modern tasks and datasets. Besides, current SOTA models almost entirely dependent on human-designed topologies and operations. So, we think human designed new concept is necessary and will still promote the research on NAS.
>
>
>   Then, we analyze the intrinsic difference between RevCol and previous methods:  the reversibility, especially for HRNet[2] and NAS-based networks (eg. Auto-DeepLab[3] and RandWire[4]). HRNet is built to perform high-resolution representation by cross resolution feature fusion and maintaining the large resolution feature maps. NAS based method such as Auto-DeepLab aims to search the optimal feature fusion path in a hyper-network and RandWire generates random wired neural networks for image recognition. All of these contain cross level feature fusion or cross level connection, but lack the key component of RevCol: reversibility.
>
>
>   HRNet, NAS based networks and other conventional networks compress the unrelated features during the IB feature learning paradigm, as shown in Fig. 1 Left in the paper. However, the unique property in RevCol, the reversible multi column connection, maintain lossless information during propagation. This property forces RevCol to learn disentangled features. Because the output of last level in last column embeds most task-relevant concepts under the IB learning principle. The reversible connection makes the last column in RevCol keeping the same informantion as pervious column for inversing. So the task-irrelevant features have to be maintained in shallow levels of last columns in a disentangled manner.  This also shown in Fig. 1 Right in the paper, the last column in RevCol embeds the task relevant concepts while maintaining the same information as the previous columns in a disentangled representation.
>
> 2. **"The shown experimental performance is competitive but the promotion over previous ones is not so evident. I admit the number is almost saturated, but any other advantage this method could bring needs to be demonstrated."**
>
> - The promotion over previous methods mainly includes:
>
>   1.  The reversibility: RevCol keeps the same information between columns that learns the disentangled representation. This prevent discarding the task irrelevant information during  pre-training, that is beneficial to downstream tasks.
>
>   2.  The scalability: RevCol opens up a new scaling-up dimension, the reversible columns, in addition to network depth and channel width. Increasing the number of reversible columns is similar to increasing both width and depth in a certain degree. We demonstrate RevCol shows high upper bound of large models even with purely convolution blocks. As shown in Tab. 2, Tab. 3 and Tab. 4 in the paper, RevCol-H achieves the best COCO detection and ADE20K segmentation result (We have updated the results in the paper) among pure (static) CNN models.
>
>   3.  The GPU memory consumption is friendly: with the propriety of reversibility, the GPU memory cost is O(1) w.r.t. model size when scaling up RevCol, as described in the submitted Appendix C.4. This reduces the hardware requirements. The training cost of large models is unaffordable without this property.
>
>   4.  This general architecture with feature disentangle may inspire some new research topics. The lossless information propagation and feature disentangle nature of RevCol may be applied to self supervised learning and low-level vision tasks. For example, the reversible column connection keeps lossless information in RevCol that could natually avoid collapsing, which is the key problem in contrastive learing that need many optimization techniques to prevent.

---

> > ### Author Response · Authors · 2022-11-18
> > **Response to Reviewer AtgT Part 2**
> >
> > 3. **"Real throughput/latency needs to be measured to more accurately validate the model budget, not just FLOPs or params. The introduced connections seem to introduce larger latency on real hardware which is not so related to FLOPs numbers."**
> > - Indeed we are aware of the current model variants of RevCol introduce large latency compared with other works of similar #params and #FLOPs. However, we did this work mainly on research purpose rather than providing an efficient substitute for ConvNeXt. And we do made an analysis for latency.
> >      We analyze the constitution of latency and get the following results. Among all the factors of latency, fragmented access of memory takes a large parts. In specific, RevCol-L consists 88 building blocks with (8, 16, 48, 16) blocks in each level. ConvNeXt-L only consists only 36 blocks. To make a fairly comparison, we construct a 88-block ConvNeXt of similar flops, *ConvNeXt-L (deep)*, then measure the latency. In Tab. 1, we can see compared to ConvNeXt-L (deep), RevCol-L introduce 23% computation overhead. Next, we further analyze the fusion module. After removing the up-sample and down-sample the connection, the overhead of RevCol-L is negligible.
> >      Note that the massive blocks design and up/down-sample connection is not a necessity for reversible and disentanglement of information (e.g. In our RevCol-ViT, which is isotropic, feature fusion can be implemented as a simple summation) as in Appendix B.
> >      We think these can be overcame by some hardware or compiler optimization (e.g. online operator fuse like JIT). Besides,
> >      if we could find a wide and shallow building block for each level, the latency would not be a problem. This is a direction for further research.
> >
> > | Model                        | #Blocks    | Latency/ms | ΔΔ     |
> > | ---------------------------- | ---------- | ---------- | ------ |
> > | ConvNeXt-L                   | 3,3,27,3   | 78.3       |        |
> > | ConvNeXt-L (deep)            | 8,16,48,16 | 100.5      | 0%     |
> > | RevCol-L                     | 8,16,48,16 | 119.9      | 19.89% |
> > | RevCol-L - upsample            | 8,16,48,16 | 111.8      | 11.79% |
> > | RevCol-L - upsample - downsample | 8,16,48,16 | 103.8      | 3.79%  |
> >
> > Table 1: Latency analysis between RevCol-L and ConvNeXt-L. Our experiments are down using an 2080ti GPU, with pytorch 1.12.0, cuda 11.3, fp16 percision. Batch size is set to 32 and the input resolution is set to 224x224.
> >
> > [1] Real E, Liang C, So D, et al. Automl-zero: Evolving machine learning algorithms from scratch[C]//International Conference on Machine Learning. PMLR, 2020: 8007-8019.
> >
> > [2] Wang J, Sun K, Cheng T, et al. Deep high-resolution representation learning for visual recognition[J]. IEEE transactions on pattern analysis and machine intelligence, 2020, 43(10): 3349-3364.
> >
> > [3] Liu C, Chen L C, Schroff F, et al. Auto-deeplab: Hierarchical neural architecture search for semantic image segmentation[C]//Proceedings of the IEEE/CVF conference on computer vision and pattern recognition. 2019: 82-92.
> >
> > [4] Xie S, Kirillov A, Girshick R, et al. Exploring randomly wired neural networks for image recognition[C]//Proceedings of the IEEE/CVF International Conference on Computer Vision. 2019: 1284-1293.

---

### Author Response · Authors · 2022-11-18
**Paper revision**

Thanks for the comments and suggestions from all reviewers. We have updated the paper in rebuttal period, the revision mainly includes:
1. We update the experiments result for RevCol-H on COCO object detection with DINO framework (63.0 box AP on test-dev set) and ADE20K segmentation with Mask2Former framework (61.0 mIoU). To our knowledge, it is the best COCO detection and ADE20k segmentation result among pure (static) CNN model.
2. We re-write parts of the method section with more clear and detailed expression.
3. We add more analysis about the difference and connection between RevCol and previous methods.

---

### Author Response · Authors · 2022-12-07
**Experiment Results Update**

Dear reviewers and audiances:

We got better results in COCO object detection and instance segmentation recently. The task-irrelevant information maintained in RevCol serves as another form of regularization, so using lighter regularization than defalut (ConvNeXt) can better tune the backbone for down-stream tasks. We further tuned the training hyper-params. With larger layerwise learning rate decay ratio and smaller stochastic depth (drop path) rate, our model perform better results. The RevCol-Base pretrained on ImageNet 22k performs **55.0 APbox** and **47.5 APmask** on COCO val set, RevCol-Large pretrained on ImageNet 22k performs **55.9 APbox** and **48.4 APmask** on COCO val set. Now **all variants of RevCol models perform better than other counterparts (ConvNeXt, Swin, RepLKNets, etc.).** Besides,  our largest model RevCol-H reaches **63.8 APbox** on COCO test-dev set, better than Swinv2-G (63.1 APbox).

If you have others questions, please feel free to raise in comments.

---

### Decision · Program_Chairs · 2023-01-20

**Decision:**

Accept: poster

**Justification For Why Not Higher Score:**

Experimental results fall short of the kind of dramatic boost over prior work that would be needed to elevate the recommendation above poster.

**Justification For Why Not Lower Score:**

The proposed architecture is interesting and results are sufficient as a proof of concept.

**Metareview: Summary, Strengths And Weaknesses:**

The paper proposes a new variant of reversible network architecture, which stacks layers into columns and connects columns in series so as to encourage gradual evolution of semantic features over the resulting two-dimensional grid of layers.  All reviewers lean toward accept and the authors provide substantial replies to reviewer questions.  The Area Chair agrees with reviewers about the overall merits of the paper.  The idea is interesting.  Experiments demonstrate performance comparable to existing architectures, providing proof the concept is at least competitive while offering the additional benefit of reversibility.

**Note From Pc:**

if the above contains the word "oral" or "spotlight" please see: "oral" presentation means -> notable-top-5% and "spotlight" means -> notable-top-25%. As stated in our emails, we are disassociating presentation type from AC recommendations

---

> ### Author Response · Authors · 2023-02-15
> **Comment on "Justification For Why Not Higher Score"**
>
> Dear ICLR 2023 Program Chairs, Area Chairs, and All Audiences,
>
> Thanks for the review and interests to our work. From the meta-review above, I noticed ***the reason for why not higher score*** states our results **fall short of the kind of dramatic boost over prior works**. I **do NOT totally agree** with this judgement. The original propose of our work is to reveal the design of reversible columns could be a general paradigm in foundation models, beyond the common idea guided by IB (Information Bottleneck) learning. In other words, the pioneering work does not aim at providing a super-fast, accurate and plug-and-play model for industrial community. That's why we choose ICLR rather than other conferences. Futhermore, we update our work several times during rebuttal period. In all down-stream tasks, RevCol model variants surpass the competitors by at least 1 point, while none of the reviewers response to these results. Reviewers also ignores the largest model, which took us a lot of pains. We boost a static, kernel 3, pure CNN model to 90% Top-1 Accuracy in ImageNet-1k classification and both 60%+ in detection and segmentation tasks. By the knowledge of us, this is merely or never seen before. Therefore, I do think the pioneering, thought-provoking RevCol is worth notice from audiences all over the world. I will attend the physical conference with our work in Rwanda.
>
>
>
> Best,
>
> Larry (Yuxuan) Cai

---

> > ### Comment · Area_Chair_t77J · 2023-02-15
> > **Re: Comment on "Justification For Why Not Higher Score"**
> >
> >
> > The Area Chair stands by statements made in the metareview regarding performance improvements over prior work.
> >
> > In apples-to-apples comparisons of the proposed RevCol model to baselines with similar parameter count and computational cost, Tables 1, 2, and 3 all show accuracy improvements of less than 1% absolute for RevCol over the closest baseline.  In particular:
> >
> >
> > Table 1:
> >
> > ImageNet-1K: RevCol-B (84.1) vs ConvNeXt-B (83.8)
> >
> > ImageNet-22K: RevCol-XL (88.2) vs ConvNeXt-XL (87.8)
> >
> >
> > Table 2:
> >
> > COCO (1K pre-train): RevCol-B (53.0 AP) vs ConvNeXt-B (52.7 AP)
> >
> > COCO (22K pre-train): RevCol-L (55.6 AP) vs ConvNeXt-L (54.8 AP)
> >
> >
> > Table 3:
> >
> > ADE20K (1K pre-train): RevCol-B (49.0 mIoU) vs RepLKNet-B (49.9 mIoU) (worse than baseline)
> >
> > ADE20K (22K pre-train): RevCol-B (53.4 mIoU) vs ConvNeXt-L (53.2 mIoU)
> >
> >
> > The 90% Top-1 accuracy on ImageNet-1K for RevCol is only observed when pre-training with an additional dataset of 168 million semi-labeled images.  No results for baseline architectures are reported in this scenario, so there is no way to separate the contribution of the proposed architecture (reversible columns) from that of additional training data.  Similarly, the 60+ results on object detection and segmentation require additional training data and no comparison is made to baselines trained in the same manner.  While it is interesting to highlight the capabilities reached with more data, any claim of advantage over other architectures in this setting would require a proper experimental baseline.